# Macro CD5L⁺ deteriorates CD8⁺T cells exhaustion and impairs combination of Gemcitabine-Oxaliplatin-Lenvatinib-anti-PD1 therapy in intrahepatic cholangiocarcinoma

Jia-Cheng Lu [1,2,3,9], Lei-Lei Wu[4,9], Yi-Ning Sun[4,9], Xiao-Yong Huang[1,2,9], Chao Gao[2,9], Xiao-Jun Guo[1,2,3], Hai-Ying Zeng[5], Xu-Dong Qu[6], Yi Chen[2], Dong Wu[7], Yan-Zi Pei[1,2,3], Xian-Long Meng[1,2,3], Yi-Min Zheng[1,2,3], Chen Liang[1,2,3], Peng-Fei Zhang[3], Jia-Bin Cai[1,2], Zhen-Bin Ding[1,2], Guo-Huan Yang[1,2], Ning Ren[1,2], Cheng Huang[1,2], Xiao-Ying Wang[1,2], Qiang Gao[1,2], Qi-Man Sun[1,2], Ying-Hong Shi[1,2], Shuang-Jian Qiu[1,2], Ai-Wu Ke[2,3], Guo-Ming Shi [1,8,9] ✉, Jian Zhou [1,2,3] ✉, Yi-Di Sun [4] ✉ & Jia Fan [1,2,3] ✉

Intratumoral immune status influences tumor therapeutic response, but it remains largely unclear how the status determines therapies for patients with intrahepatic cholangiocarcinoma. Here, we examine the single-cell transcriptional and TCR profiles of 18 tumor tissues pre- and post- therapy of gemcitabine plus oxaliplatin, in combination with lenvatinib and anti-PD1 antibody for intrahepatic cholangiocarcinoma. We find that high CD8 GZMB⁺ and CD8 proliferating proportions and a low Macro CD5L⁺ proportion predict good response to the therapy. In patients with a poor response, the CD8 GZMB⁺ and CD8 proliferating proportions are increased, but the CD8 GZMK⁺ proportion is decreased after the therapy. Transition of CD8 proliferating and CD8 GZMB⁺ to CD8 GZMK⁺ facilitates good response to the therapy, while Macro CD5L⁺−CD8 GZMB⁺ crosstalk impairs the response by increasing CTLA4 in CD8 GZMB⁺. Anti-CTLA4 antibody reverses resistance of the therapy in intrahepatic cholangiocarcinoma. Our data provide a resource for predicting response of the combination therapy and highlight the importance of CD8⁺T-cell status conversion and exhaustion induced by Macro CD5L⁺ in influencing the response, suggesting future avenues for cancer treatment optimization.

Intrahepatic cholangiocarcinoma (iCCA) is a high aggressive neoplasm originating from the epithelium of the intrahepatic biliary tree and ranks as the second most common primary liver cancer after hepatocellular carcinoma (HCC)[1]. The incidence of iCCA has increased dramatically over the past 30 years. In high-income countries, such as the UK and the USA, the incidence of iCCA has consistently and steadily increased (from 0.1 cases per 100,000 to 0.6 per 100,000). In Thailand and China, the incidence is up to 40 times higher. The prognosis of iCCA remains dismal because patients often present with advanced stage at the initial visit, largely due to the highly invasive nature of iCCA and the lack of effective treatment[1–4]. Gemcitabine-based chemotherapy, including gemcitabine plus cisplatin (GemCis) and gemcitabine

A full list of affiliations appears at the end of the paper. ✉e-mail: shi.guoming@zs-hospital.sh.cn; zhou.jian@zs-hospital.sh.cn; ydsun@ion.ac.cn; fan.jia@zs-hospital.sh.cn

plus oxaliplatin (GEMOX), is the first-line treatment for patients with advanced iCCA. Unfortunately, low objective response rates (ORRs) limit the benefits in iCCA patients[5]. Despite the encouraging results of targeted therapy and immune checkpoint inhibitor (ICI) therapy achieved in patients with some malignancies, the majority of iCCA patients still gain limited benefits from single-agent treatment due to the low frequency of targetable mutations and a lack of microsatellite instability. Fibroblast growth factor receptor (FGFR) signaling and/or the programmed death 1 (PD1)/ programmed death ligand-1 (PD-L1) pathway are abnormally activated in some patients with iCCA, which prompted calls for targeted therapy at FGFR signaling and/or immune checkpoint blockade (ICB). In practice, combination therapies have become an increasingly popular strategy to improve therapeutic efficacy in malignancies, especially combinations of chemotherapy, targeted therapy, and ICB[6,7].

Chemotherapy-based systemic therapy in combination with tyrosine kinase inhibitors (TKIs) and/or anti-PD1/PD-L1 antibodies has shown promising results for advanced iCCA[2,8]. Lately, TOPAZ-1 trial showed GemCis plus Durvalumab significantly improved overall survival (OS) of patients with advanced biliary tract cancer (BTC) compared to GemCis chemotherapy (median OS: 12.8 months versus 11.5 months)[9]. Our group reported the results from a phase-II clinical trial (NCT03951597) and showed combination therapy (Gemcitabine, Oxaliplatin, Lenvatinib, and anti-PD1 antibody, GOLP) as first-line therapy for patients with advanced iCCA had a promising ORR of 80% and median OS of 22.5 months[10,11]. Thus, this regimen was recommended as first-line therapy for advanced iCCA by the Chinese Society of Clinical Oncology (CSCO) in 2021[12]. Although our results showed DNA damage response (DDR) related mutations were related to tumor response[11], the determinants underlying the effective responses to GOLP in patients with iCCA remain largely unclear.

Single-cell sequencing holds several technical advantages for understanding the complex profiles of the tumor immune microenvironment[13,14] and has been used to uncover the determinants of anti-PD1 treatment response in melanoma, breast cancer, and clear cell renal cell carcinoma[15–18]. Recently, several studies have generated single-cell atlases of HCC and iCCA[19,20]. We also deciphered the distinct immune ecosystems of HCC cases with different relapse stages[21]. Ma et al. also reported a single-cell atlas reflecting tumor cell evolution with therapy in HCC and iCCA[22]. These studies have provided valuable resources, yet the exploration to clarify the determinants of the outcome of combination treatment in iCCA is urgent. In addition, chemotherapy agents and/or targeted drugs have been shown to remodel tumor microenvironment (TME) and enhance sensitivity to anti-PD1 therapy in tumors[23–26]. Therefore, it is necessary to systematically assess the dynamic changes that occur in the TME of iCCA with the promising GOLP treatment.

In this work, we utilize single-cell RNA and T-cell receptor (TCR) sequencing to characterize the cellular and molecular dynamic atlas of the TME in patients with iCCA treated with the GOLP regimen. We identify the cell types responsible for sensitivity to GOLP treatment and the key cellular subtypes predicting therapeutic response and uncover the underlying mechanism of GOLP therapy in iCCA. Our study provides a comprehensive and dynamic cell atlas of iCCA patients treated with the GOLP regimen and lays a theoretical basis for personalized and optimized therapy in iCCA.

## Results

### The GOLP regimen showed high therapeutic efficacy in iCCA
We retrieved the public data of 33 clinical trials of first-line treatment of BTC including iCCA since 2010 (https://clinicaltrials.gov/) and summarized their ORR, median progression-free survival (mPFS) and mOS (Supplementary Data 1). In the 14 clinical trials, patients with iCCA achieved an ORR of 13.5–80% with different strategies (Supplementary Data 1). Our previous study[10,11] showed that the GOLP regimen achieved

the best ORR (80%) in the treatment of advanced iCCA, while lenvatinib plus anti-PD1 antibody or GEMOX showed ORRs of approximately 30% in the treatment of advanced iCCA (Fig. 1A).

We also analyzed the therapeutic efficacy of the GOLP regimen in 28 iCCA patients from a real-world study (FDU-ZS-iCCA-T cohort). These patients were received three cycles of GOLP (Fig. 1B; **Methods**), and tumor responses were evaluated per Response Evaluation Criteria in Solid Tumors version 1.1 (RECIST 1.1) by an independent radiological review committee (IRRC). The resectability of each subject was discussed by a multidisciplinary team (MDT), and radical resection was performed after three cycles of GOLP treatment (Fig. 1B). Among the 28 iCCA patients, 16 achieved partial response (PR) and 12 had stable disease (SD) at the end of three cycles of GOLP (Fig. 1C, D). H&E staining validated different degrees of pathological remission after GOLP treatment (Supplementary Fig. 1A). These results demonstrate that GOLP is an effective strategy for patients with iCCA in clinical practice.

To further validate whether triple-strategies combination therapy GOLP showed better therapeutic efficacy than one or two combinations, we also used two mouse iCCA cell lines (mIC-22 constructed by our team and AY-LTC2 provided by Prof. Yongzhong Liu as a gift) (Supplementary Fig. 1B–F; **Methods**) to construct the tumor-bearing CL57BL/6 mice, and treated with different drug combinations (Fig. 1E). After three cycles of treatment, mimic GOLP group showed more obvious tumor shrinkage than other therapeutic groups (Fig. 1F and Supplementary Fig. 1L), suggesting the GOLP regimen showed high therapeutic efficacy in iCCA.

### Single-cell profiling revealed dynamic changes after GOLP treatment in iCCA
To explore the underlying mechanism of GOLP treatment in patients with iCCA, we performed single-cell RNA and TCR sequencing of 8 paired pretreatment/post-3 cycles of GOLP treatment samples (**Discovery cohort**), and 2 treatment-naive samples before operation (Supplementary Fig. 1I, J). Principal component analysis showed no apparent different clusters between pre-treatment biopsy samples and 2 surgery samples without preoperative treatment, indicating these biopsy samples and the surgery samples are comparable (Supplementary Fig. 1K). We also collected 55 tumor samples from patients with advanced iCCA before GOLP treatment, including 27 patients from NCT03951597 and 28 patients from a real-world study (FDU-ZS-iCCA-T) as validation cohorts (Fig. 2A and Supplementary Fig. 1I; Supplementary Data 1). A recent study showed that pathological subtype (large-duct and small-duct type) may affect the treatment options and outcome of patients with iCCA[1]. However, no significant difference was observed in the response to GOLP treatment between the two subtypes of iCCA (Chi-sqaure test, two-sided, $P = 0.42$; Supplementary Data 1).

From our scRNA-seq data, a total of 131,139 cells were clustered into 12 cell types with the uniform manifold approximation and projection (UMAP) method (Fig. 2B and Supplementary Fig. 2A; Supplementary Data 2; **Methods**) after quality control. The immune cell types included T cells (*CD3D*, *CD3E*, and *CD3G*; 39.1%), myeloid cells (*CD14*, *FCGR3A*, and *LYZ*; 15.9%), natural killer (NK) cells (*KLRD1*, *GNLY*, and *NKG7*; 13.1%), B cells (*MS4A1*, *CD79A*, and *CD19*; 11.2%), neutrophils (*S100A8* and *S100A9*; 2.33%), plasma cells (*MZB1* and *CD38*; 1.26%), plasmacytoid dendritic cells (pDCs, *LILRA4*, *TCF4* and *BCL11A*; 0.57%), and mast cells (*TPSB2* and *CD63*; 0.35%). The nonimmune cell types included epithelial cells (*KRT18*, *EPCAM*, and *CLDN4*; 11.0%), fibroblasts (*FN1*, *COL1A1* and *DCN*; 2.99%), endothelial cells (*PECAM1*, *CDH5*, and *TM4SF1*; 1.98%), and hepatocytes (*APOC3*, *TTR* and *ALB*; 0.28%) (Fig. 2C and Supplementary Fig. 2B, C). Cell type to cell type correlation matrix showed every annotated cellular phenotype was highly correlated with itself, indicating the uniqueness of the expression pattern of the cellular phenotype (Fig. 2D). A higher percentage of immune cells (T cells,

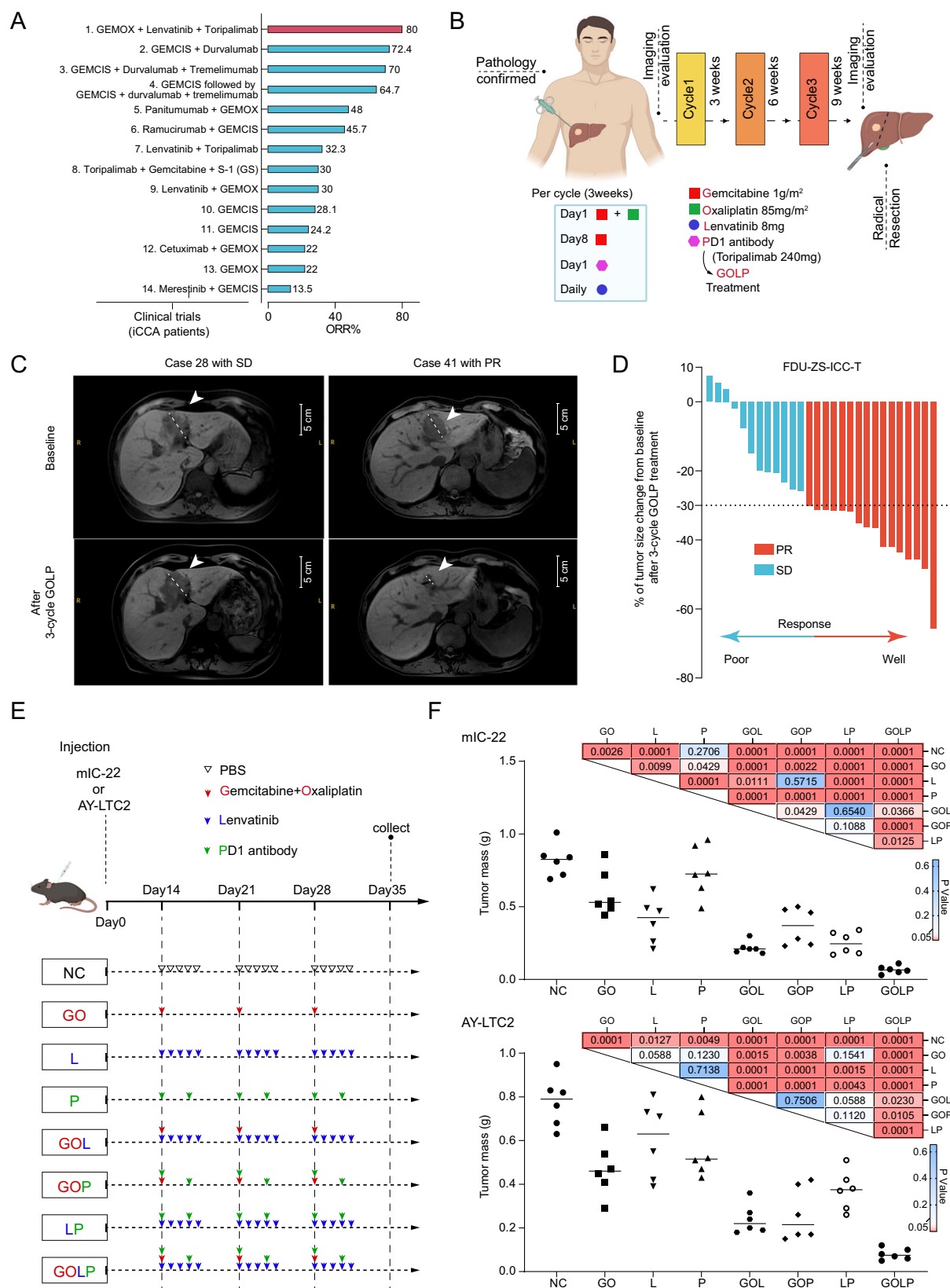

NK cells, pDCs, neutrophils, B cells, and plasma cells) than nonimmune cells (fibroblasts, epithelial cells, endothelial cells, and hepatocytes) was observed in all the samples (Fig. 2E). To investigate the dynamic changes in major cell populations after GOLP treatment, we calculated the post-to-pre ratio of major cell types upon GOLP treatment in paired tumor samples (Pt. 1-Pt. 8). Interestingly, T cells, myeloid cells and fibroblasts were increased after GOLP treatment (Fig. 2F).

To further explore the correlation between cell populations and tumor response to GOLP treatment, we employed two indicators, predictive index (Pi) and therapeutic index (Ti), as previously

**Fig. 1 | The GOLP regimen showed high therapeutic efficacy in iCCA. A** Bar plot showing the overall response rate (ORR) in clinical trials of different first-line therapeutic strategies for iCCA patients. **B** Workflow of combination therapy of Gemcitabine, Oxaliplatin, Lenvatinib, and anti-PD1 antibody (GOLP) for iCCA patients. Created with BioRender.com. **C** Representative results of 'stable disease' (SD) and 'partial response' (PR) iCCA patients at baseline and after 3 cycles of GOLP treatment. SD and PR were evaluated according to the Response Evaluation Criteria in Solid Tumors version 1.1 (RECIST1.1). The arrows indicate the tumors of interest.

**D** Percentage of tumor size reduction after 3 cycles of GOLP treatment in FDU-ZS-ICC-T cohort. Patients evaluated as SD were defined as 'Poor response' and the patients evaluated as PR were defined as 'Well response'. **E** Workflow of GOLP and other control groups for the treatment of iCCA-bearing mouse with mIC-22 or AY-LTC2. Created with BioRender.com. **F** Tumor mass after three cycles of different treatments (t test, two-sided). N= 6 biologically independent animals for each group. Source data are provided as a Source Data file.

described[18]. Pi and Ti were used to measure the correlations of changes in tumor size with cell type proportions at baseline and the alteration of cell type proportions after GOLP treatment, respectively (**Methods**). A positive Pi or Ti indicated a favorable predictor of GOLP response, where a higher proportion of the specific cell type at baseline or an increase in the proportion of the cell type during treatment was correlated with a higher degree of tumor shrinkage. Upon Pi and Ti analysis, we found that high proportions of plasma cells, fibroblasts, and T cells at baseline were associated with a favorable response to GOLP treatment, whereas high proportions of myeloid and NK cells at baseline were associated with an unfavorable response to GOLP (Fig. 2G, H).

### Four meta-clusters of tumor cells presented distinct responses to GOLP treatment

Most nonimmune cells in iCCA tumors were epithelial-derived tumor cells with high expression of *KRT19* and *EPCAM* (Fig. 2B, C). Tumor cells were confirmed by high copy-number variations (CNVs) inferred from scRNA-seq expression profiles (Supplementary Fig. 3A, B), consistent with the amplification characteristic of solid tumors[27]. In addition, our inferred CNV variations were highly consistent with a previous reported pattern of chromosomal aberrations in cholangiocarcinoma such as chromosome 3p, 6q, 13q deletions, and chromosome 1q amplification[28] (Supplementary Fig. 3A). To reveal the changes in tumor cells upon GOLP treatment, we performed functional enrichment analysis of differentially expressed genes between post- and pre-GOLP treatment tumors. Metabolic and catabolic pathways were enriched in the pre-GOLP treatment tumor cells, whereas inflammatory response, cell activation and adaptive immune response-associated pathways were enriched in tumor cells after GOLP treatment (Fig. 3A), suggesting a shift from a metabolism-activated state to an immune-activated state upon GOLP treatment. In addition, higher expression of *CD274*, *HLA-A*, *PCNA*, *FGF11*, and *FGF18* in tumor cells before treatment was a favorable predictor of tumor shrinkage (Fig. 3B; Supplementary Data 3), and gene set enrichment analysis (GSEA) showed that PD1 signaling, positive regulation of vasculature development and signaling by KIT in disease were enriched in the tumor cells from patients with a good response to GOLP therapy (Fig. 3C). Tumor cells were clustered by individual sample (Fig. 3D), indicating obvious tumor heterogeneity in iCCAs as described in previous reports[22]. These results of high tumor heterogeneity of the TME also revealed combination therapy for iCCA patients as a rational option.

Over 3000 genes were preferentially expressed in individual samples (Supplementary Fig. 3C). We then used the consensus non-negative matrix factorization (cNMF) method to obtain consensus clustering of tumor cells on the basis of common expression patterns[29] and identified 4 meta-clusters (C1-C4) with distinct gene expression profiles (Fig. 3E and Supplementary Fig. 3D; **Methods**). Enrichment analysis of the genes specifically expressed in each meta-cluster suggested their corresponding functions: immune response and antigen processing & presentation for C1, ATP metabolic process for C2, cellular detoxification and negative regulation of inflammatory response for C3, and activator protein 1 (AP-1) pathway for C4 (Fig. 3E and Supplementary Fig. 3E; Supplementary Data 3). In particular, we also identified *HLA-DRA* and *HLA-DRB1* (enriched in C1)

and *NQO1* and *GPX2* (enriched in C3) as showing higher expression levels in the PR group (Fig. 3F). We calculated the C1-C4 scores for each patient in the GOLP-treated cohort and performed Pi analysis for each tumor cluster. The results showed that a high C1 or C3 score was associated with a good response to GOLP treatment (Fig. 3G). Based on the expression levels of marker genes of four meta-clusters in the scRNA data, we could also classify a retrospective treatment-naive iCCA cohort (FU-iCCA) into 4 subgroups (Fig. 3H; **Methods**). These results indicated the four meta-clusters calculated by our scRNA data also exhibited robustness in a larger iCCA cohorts with bulkRNA data, and the patients with high cluster C1/C3 probably receive benefit from GOLP therapy.

### Tumor with high proportion of Macro CD5L⁺ was insensitive to GOLP therapy

Myeloid cells are considered critical mediators in the TME and can modulate T-cell responses in tumor immunity[30]. We next explored the alteration in gene expression profiles of myeloid cells. We found that the enriched genes in myeloid cells from tumor sample at baseline were significantly associated with innate immune response and inflammatory response (Fig. 4A; Supplementary Data 4). Moreover, genes upregulated in the myeloid cells of patients with poor response to GOLP (evaluated as SD) were mainly involved in negative regulation of cell differentiation and immune system process. However, in post-GOLP treatment samples, enriched genes in myeloid cells were related to adaptive immune response, and highly expressed genes in the well response to GOLP therapy (evaluated as PR) largely involved in inflammatory response and positive regulation of leukocyte activation pathways (Fig. 4B; Supplementary Data 4), supporting myeloid cells play an important role in mediating responses to GOLP treatment.

To provide deeper insights into the myeloid compartment in iCCA, we performed unsupervised graph-based clustering of myeloid cells and identified 3 major myeloid types, macrophages (*CD68*, *CD163*, and *CSTB*), monocytes (*CD14*, *FCN1*, and *IL1B*), and dendritic cells (DCs; *CLEC1A*, *CD1C*, and *CD1E*), based on the expression of canonical cell markers (Supplementary Fig. 4A, B). Based on the expression of cluster-specific marker genes, we further identified seven subsets of macrophages (Macro GPNMB⁺, Macro CD5L⁺, Macro SPINK1⁺, Macro MRC1⁺, Macro IGLC2⁺, Macro IL32⁺, and Macro FBP1⁺), four monocyte subtypes (Mono CCL3⁺, Mono CXCL10⁺, Mono CCL20⁺, and Mono FCN1⁺), and three DC subtypes (DC IDO1⁺, DC CD1C⁺, and DC TYMS⁺) (Fig. 4C, D). Although we found that myeloid cells were significantly expanded in the TME of post-GOLP treatment iCCA samples (Fig. 4E), most subtypes enriched in post-GOLP samples showed high patient occupancy (Fig. 4F; **Methods**). For instance, most Mono CCL20⁺ and Macro FBP1⁺ in post-GOLP samples were from Pt. 2 and Pt. 1, respectively, indicating that myeloid cells showed sample-specific characteristics and drastic changes during GOLP treatment (Fig. 4G, Supplementary Fig. 4C, D). The characteristics of high patient occupancy were only observed in tumor cell and myeloid types, but not in T-cell types (Supplementary Fig. 4E), consistent with the plasticity of myeloid cells reported before[31]. Among these myeloid subclusters, we found that the proportion of DC IDO1⁺ (P = 0.019) was significantly decreased after GOLP treatment (Supplementary Fig. 4F), probably releasing the brake of immune inhibition[32–34]. Notably, a high proportion of Macro CD5L⁺ cells at baseline was significantly associated with

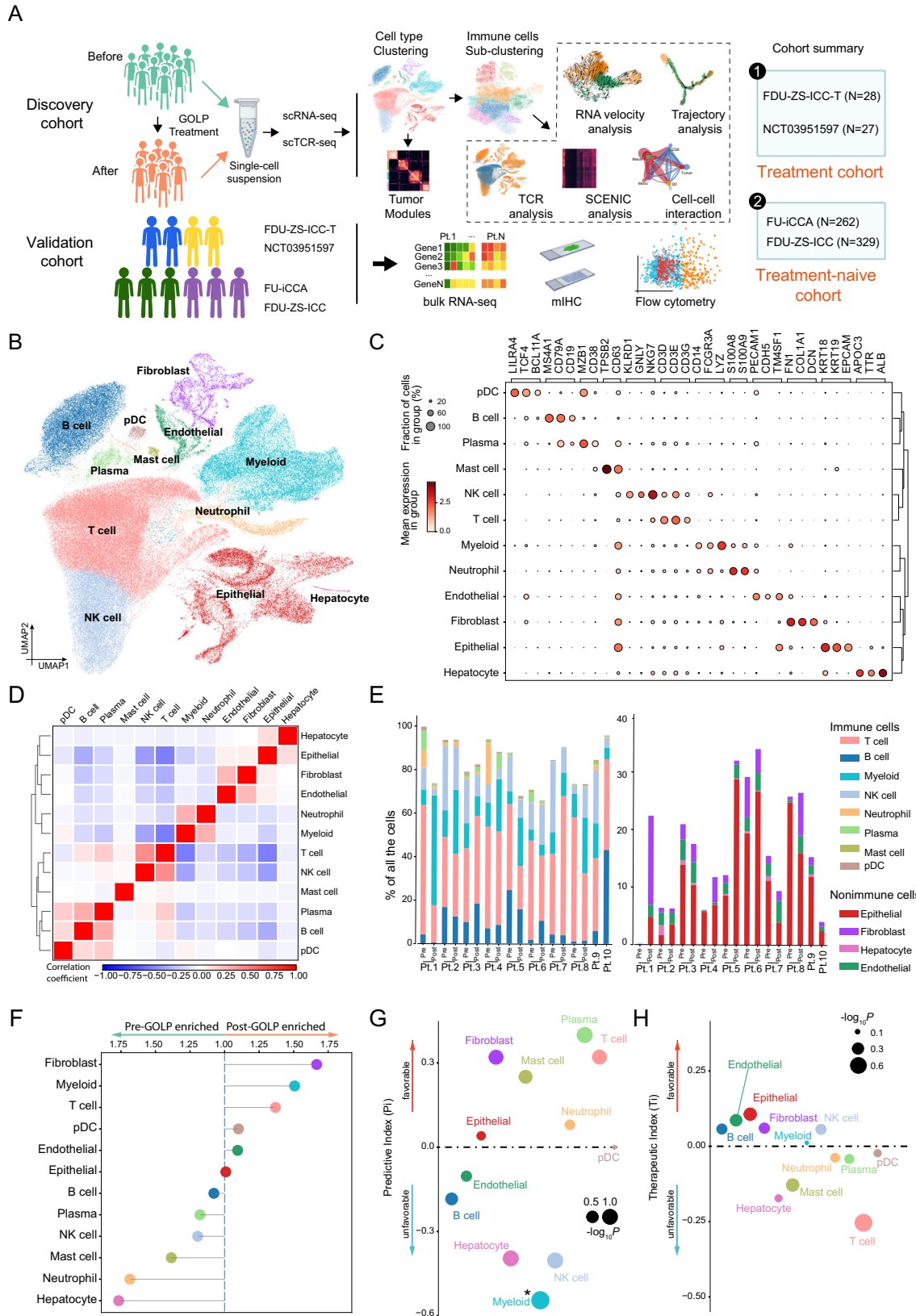

poor response to GOLP treatment (Fig. 4H and Supplementary Fig. 4G).

We further explored the predictive value of Macro CD5L⁺ in two cohorts of iCCA patients treated with GOLP (NCT03951597 and FDU-ZS-iCCA-T) by multiplex immunohistochemistry (mIHC). The results showed that Macro CD5L⁺ cells were much more abundant in tumor

samples of patients with poor response to the GOLP regimen than those of patients with good response to the GOLP regimen (Fig. 4I, J and Supplementary Fig. 4H). In addition, using gene signatures of functionally defined myeloid subtypes[21,35,36], we found that Macro CD5L⁺ exhibited higher M2, anti-inflammatory and phagocytosis signatures (Fig. 4K; Supplementary Data 4). This observation was

**Fig. 2 | Single-cell profiling of the iCCA microenvironment during GOLP treatment. A** Study diagram and experimental workflow. Created with BioRender.com. **B** Uniform manifold approximation and projection (UMAP) plot depicting 12 cell lineages from the 18 samples (N = 131,139 cells). **C** Dot plot showing the average expression levels of marker genes across the 12 cell types. The dot size indicates the fraction of cells expressing the marker gene in each cluster, and the dot color represents the average expression level of the marker gene in each cluster. The dendrogram on the right represents the hierarchical clustering of different cell types based on the expression levels of the indicated marker genes. **D** Heatmap showing the relationship among the 12 cell types based on the correlation of gene expression between each pair of cell types as assessed by Pearson's correlation analysis. Blue and red indicate negative and positive correlations, respectively. **E** Bar plots show the distribution of immune and nonimmune cells in each patient. The samples are presented in pre- and post-GOLP treatment pairs for the 8 patients (Pts. 1–8), and 2 patients underwent surgery without pre-GOLP treatment (Pts. 9 and 10). Pt., patient. For every sample, annotated cell types were divided into immune cell type (T cells, NK cells, pDCs, neutrophils, B cells, and plasma cells: Left) and non-immune cell type (fibroblasts, epithelial cells, endothelial cells, and hepatocytes: Right). For every sample, the percentage of immune cell type plus percentage of non-immune cell type is 100%. **F** The enrichment scores of different cell types in the pre- and post-GOLP treatment groups. The colors represent different cell types. **G, H** The Pi (**G**) and Ti (**H**) values of major cell types in iCCA patients treated with GOLP (N = 8 paired samples). Pi, predictive index; Ti, therapeutic index. The dot size represents the significance evaluated as -log$_{10}$ (*P* value). *: *P* < 0.05, Wald test, two-sided. Source data are provided as a Source Data file.

consistent with a previous report that CD5L$^+$ can induce an anti-inflammatory cytokine profile and promote M2 polarization[37]. Additionally, we examined the expression of known MDSC markers *S100A8*, *S100A9* and *S100A12*[38] and found them barely expressed in Macro CD5L$^+$ cells (Supplementary Fig. 4H), excluding the possibility of Macro CD5L$^+$ as a group of MDSC. Taken together, these results indicate that Macro CD5L$^+$ proportion at baseline can predict the response to GOLP treatment for iCCA.

## High baseline levels of CD8 GZMB$^+$ predicted a favorable response to GOLP treatment

We identified 12 subsets of conventional (NK, CD4$^+$ and CD8$^+$) and unconventional (γδT) lymphoid cells (Supplementary Fig. 5A), including two NK cell subtypes (NK GNLY$^+$ and NK FCER1G$^+$), four CD4 T cell subtypes (CD4 naive, CD4 CXCL13$^+$, CD4 SOCS3$^+$ and CD4 Treg), five CD8$^+$ T cell subtypes (CD8 proliferating, CD8 GZMK$^+$, CD8 GZMB$^+$, CD8 KLRB1$^+$ and CD8 Trm) and γδT cells (Fig. 5A, B; Supplementary Data 5). Cell type to cell type correlation matrix showed every annotated cellular phenotype was highly correlated with itself, indicating the uniqueness of expression pattern of the cellular phenotype (Supplementary Fig. 5B). The proportions of CD4 SOCS3$^+$ and CD4 CXCL13$^+$ were significantly increased after GOLP treatment, while that of CD4 naive was decreased (Fig. 5C and Supplementary Fig. 5C). CD4 CXCL13$^+$ has been reported to be a tumor-activated CD4$^+$ T subtype[18], indicating a shift of CD4$^+$ T-cell populations from the naive to the activated state with GOLP treatment. Based on Pi analysis, we found that CD4 CXCL13$^+$ showed a significantly positive Pi (*P* = 0.047; Fig. 5D), consistent with the predictive role of CD4 CXCL13$^+$ in PD-L1 blockade treatment for triple-negative breast cancer (TNBC)[18]. Interestingly, CD4 CXCL13$^+$ showed the highest expression levels of PDCD1 among the CD4$^+$ T subtypes (Fig. 5E), hinting that the predictive value of CD4 CXCL13$^+$ in GOLP treatment for iCCA might be partially explained by the inhibition of PDCD1 by anti-PD1 antibody. In contrast, CD4 Treg with high expression of *FOXP3*, *BATF* and *TIGIT* showed a negative Ti (*P* = 0.033; Fig. 5D), consistent with the notion that an increase in immune-suppressive T cells leads to a poor response to GOLP therapy.

Pi analysis of the predictive value of CD8$^+$T-cell subtypes revealed high baseline levels of CD8 proliferating (*P* = 0.0006) and CD8 GZMB$^+$ (*P* = 0.016) as predictors of favorable GOLP response (Fig. 5F and Supplementary Fig. 5D). CD8 proliferating and CD8 GZMB$^+$ exhibited higher *PDCD1* expression (Supplementary Fig. 5E). CD8 proliferating have also been reported to have a role in predicting anti-PD1 therapy response in lung cancer patients[39]. Ti analysis showed that an increase of CD8 GZMK$^+$ in tumor samples was a predictor of favorable GOLP response (Fig. 5F), and a higher post-to-pre GOLP ratio of CD8 GZMK$^+$ was significantly correlated with more shrinkage of tumor (R = 0.92, *P* = 0.001; Fig. 5G). Moreover, we found that iCCA patients with high expression levels of GZMK showed a significantly better prognosis than those with low GZMK expression levels in the two large treatment-naive validation cohorts (Fig. 5H). These results suggested that CD8 GZMK$^+$ cells play a protective role in patients with iCCA. Flow cytometry analysis also showed that the baseline proportions of CD8 GZMB$^+$ and CD8 proliferating in the TME were significantly higher in patients with well response to GOLP treatment (Fig. 5I, J and Supplementary Fig. 5F), confirming their value in predicting a favorable response. In addition, mIHC analyzes further demonstrated higher baseline levels of CD8 GZMB$^+$ in the well response groups of the three cohorts treated with GOLP (Fig. 5K, L and Supplementary Fig. 5G). The results indicate that a high baseline level of CD8 GZMB$^+$ is a predictor of favorable response to GOLP treatment.

## Dynamic changes in CD8 proliferating, CD8 GZMB$^+$, and CD8 GZMK$^+$ upon GOLP treatment

Analysis of TCR variable regions has uncovered clonal expansion of CD4$^+$ or CD8$^+$ T cells with exhaustion phenotypes due to persistent tumor antigen stimulation[18,40,41]. Here, we explored the dynamic changes in T-cell clonotypes upon GOLP treatment. Our data revealed the concordance of TCR clonotypes and T-cell phenotypes (Supplementary Fig. 6A, B). The percentage of shared CD8$^+$ T-cell clonotypes was evidently increased in seven out of eight patients after GOLP treatment (Supplementary Fig. 6C). In addition, the percentages of shared clonotypes among different CD8$^+$ T-cell subtypes were significantly higher than those among different CD4$^+$ T-cell subtypes in both pre- and post-treatment samples (Supplementary Fig. 6C), indicating the clonal expansion activity of CD8$^+$ T cells upon GOLP treatment. To further demonstrate the alteration of clonotypes upon GOLP treatment, we compared the T-cell clone abundances in paired samples and identified 318 T-cell clonotypes with statistically significant differences. The clonotypes with higher abundances or presentation after GOLP treatment were designated as expanded or novel TCR clones, and those with reduced abundances in post-GOLP samples were considered as contracted TCR clones (Fig. 6A). Interestingly, T cells with expanded and contracted clonotypes upon GOLP treatment expressed exhaustion (*HAVCR2*, *TIGIT*, *LAG3*, and *ENTPD1*) and activation (*IFNG* and *TNFRSF9*) marker genes, respectively (Supplementary Fig. 6D), consistent with the characteristics of tumor-reactive CD8$^+$ T cells as previously reported[40].

Importantly, patients with high clonally expanded CD8$^+$ T cells (CD8$^+$ T Expanded) had a favorable response to GOLP treatment, while those with high clonally contracted CD8$^+$ T cells (CD8$^+$ T Contracted) presented with a poor response to GOLP treatment (Fig. 6B). These results were consistent with the relationship of CD8$^+$ T-cell clonal expansion with positive response to anti-PD1 therapy in TNBC[16]. In addition, we found that the CD8 GZMB$^+$, CD8 GZMK$^+$ and CD8 proliferating subtypes accounted for the majority of the expanded, novel and contracted clonotypes among all the T-cell subtypes (Fig. 6C and Supplementary Fig. 6E). In addition, clonally expanded CD8 GZMB$^+$, CD8 GZMK$^+$ and CD8 proliferating cells were significantly increased after GOLP treatment (Fig. 6D). These results suggested that the three cell types were the most active subtypes in response to GOLP

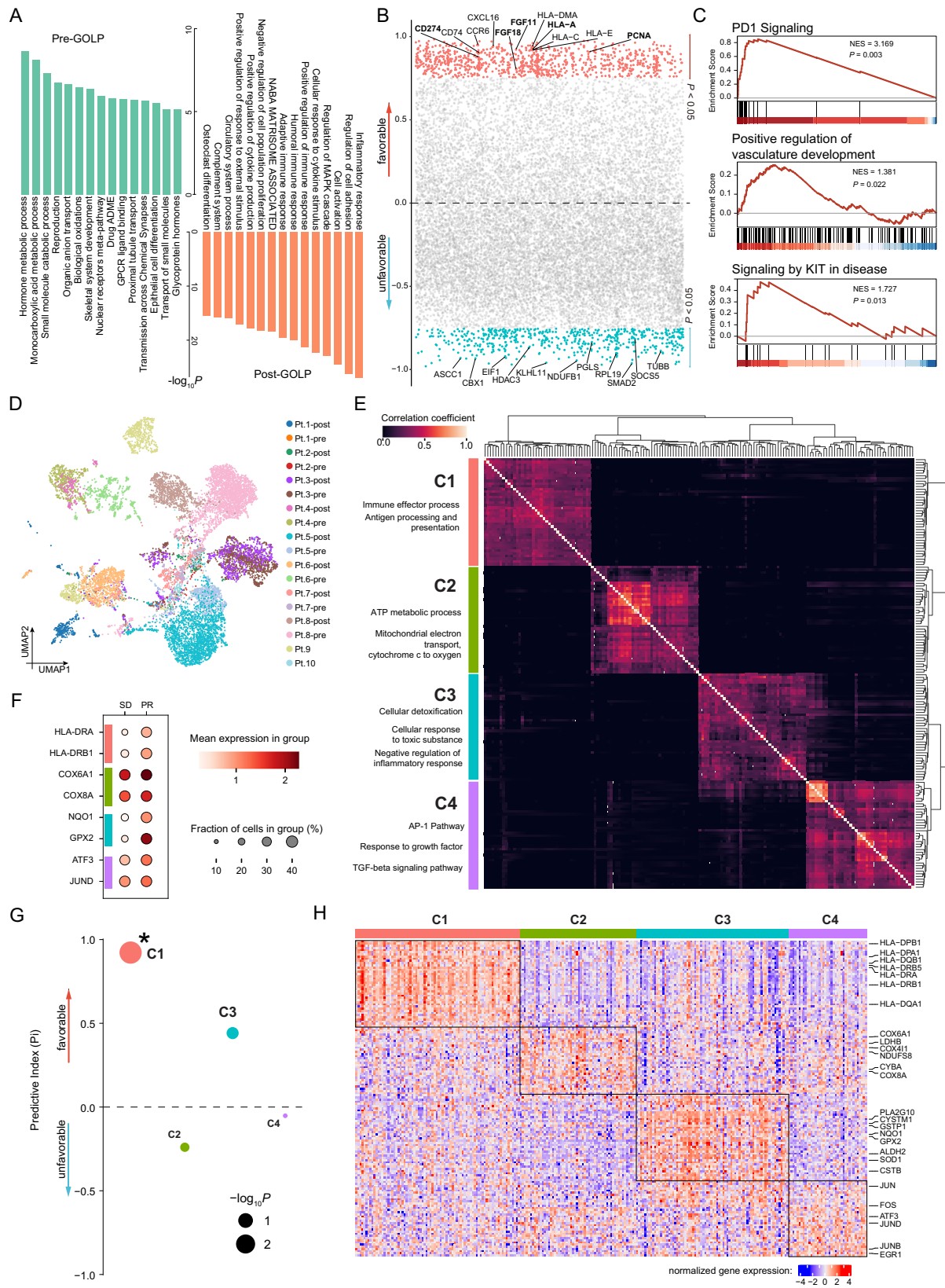

treatment. We assessed the most abundant cell type with clonal expansion in response to GOLP treatment, and found the exhaustion scores of CD8 GZMB⁺ were significantly decreased after GOLP treatment (Fig. 6E and Supplementary Fig. 6F), suggesting that GOLP treatment reshaped immune-exhausted tumor environments.

Furthermore, we confirmed that there was a higher increase in CD8 proliferating and CD8 GZMB⁺ in SD patients and a higher increase in CD8 GZMK⁺ cells in well response patients upon GOLP treatment in our validation cohorts (NCT03951597 and FDU-ZS-iCCA-T) by mIHC and flow cytometry (Fig. 6F, G).

**Fig. 3 | Tumor features correlated with iCCA prognosis and GOLP response.**
**A** Bar plot showing the top enriched pathways of differentially expressed genes between pre- and post-GOLP tumor cells (N = 18 samples). A hypergeometric test was used. A Benjamini–Hochberg adjusted $P < 0.05$ (two-sided) was used as a significance cutoff. **B** Correlation between gene features of tumor cells before GOLP treatment and tumor shrinkage (N = 8 patients). Gene features above zero were favorably associated with tumor shrinkage ($P < 0.05$: red); gene features below zero were unfavorably associated with tumor shrinkage ($P < 0.05$: blue); Pearson's correlation test (two-sided); exact $P$ values were listed in Supplementary Data 3.
**C** GSEA showed that PD1 signaling, positive regulation of vasculature development and signaling by KIT in disease were enriched in the pre-GOLP tumor cells in samples with a good response. Empirical phenotype-based permutation test (one-sided). **D** UMAP plot showing the expression profiles of tumor cells from the

18 samples (N = 14,411 cells). **E** Heatmap showing the 4 meta-clusters (C1-C4) of tumor cells identified using consensus non-negative matrix factorization (cNMF). The terms on the left represent the pathways enriched in each meta-cluster. **F** Dot plot showing the average expression levels of representative effector genes in the 4 meta-clusters between the PR and SD groups (N = 8 patients). The dot size indicates the fraction of cells expressing the specific gene in the group, and the dot color represents the average gene expression level in the group. **G** Pi values of the 4 meta-clusters (C1-C4). N = 8 patients. The dot size represents the significance evaluated as -$\log_{10}$ ($P$ value). *: $P < 0.05$. Wald test, two-sided. $P$ values of C1, C2, C3, and C4 are 0.0011, 0.57, 0.27 and 0.9, respectively. (**H**) Heatmap showing the expression profiles of the 4 meta-clusters in the large GOLP treatment-naive iCCA cohort (FU-iCCA, N = 262) with bulk RNA-seq data. The representative genes of each meta-cluster are highlighted on the right. Source data are provided as a Source Data file.

## Transition of CD8 proliferating and CD8 GZMB⁺ into CD8 GZMK⁺ influenced the response to GOLP

As CD8 GZMB⁺, CD8 GZMK⁺ and CD8 proliferating were the most active CD8⁺ subtypes in response to GOLP treatment, we next explored how these three cell types influenced response to GOLP treatment. Interestingly, we found that the clonal similarities among the three clusters (CD8 GZMB⁺, CD8 GZMK⁺ and CD8 proliferating) were much higher than those of the other T-cell subtypes (Fig. 6H and Supplementary Fig. 6G), suggesting similar origins of the three clusters[42,43]. Therefore, we next performed cell trajectory analysis using Monocle2[44] and found that CD8⁺ subtypes were allocated into four functional phases with different transcriptional states (Supplementary Fig. 7A, B). CD8 proliferating were dominantly distributed into phase I (trajectory root) with high expression of cell cycle- and mitosis-related genes, while phase III was characterized by high expression levels of genes associated with adaptive immune responses (*TIGIT*, *PDCD1*, *HAVCR2*, *S100A4*, and *CD74*; Supplementary Fig. 7B; Supplementary Data 6). We noticed that CD8 GZMK⁺ were enriched at the terminal states of the trajectory, but CD8 GZMB⁺ were dispersed from phase II to IV (Supplementary Fig. 7A, B), suggesting an intermediate status of the CD8 GZMB⁺ subtype. Consistently, RNA velocity analysis supported a transition from CD8 proliferating to CD8 GZMK⁺ through CD8 GZMB⁺ (Fig. 6I and Supplementary Fig. 7C). Additionally, TCR diversity has been reported to reflect the state (naive or terminally differentiated) of CD8⁺ T cells[16]. Our results showed that CD8 proliferating had the highest TCR diversity among the three CD8⁺ clusters in pretreatment samples (Supplementary Fig. 7D), supporting that CD8 proliferating are in a more naive state than CD8 GZMB⁺ or CD8 GZMK⁺.

Moreover, we also found that these three CD8⁺ T-cell subtypes showed distinct cytotoxic and exhausted states. CD8 GZMK⁺ had the highest cytotoxic score and the lowest exhausted score. CD8 proliferating showed a relatively high exhausted score and low cytotoxic score, while CD8 GZMB⁺ showed a diverse distribution of cytotoxic and exhausted scores (Fig. 6J; **Methods**), consistent with the intermediate state inferred from the trajectory and RNA velocity analyzes (Fig. 6I and Supplementary Fig. 7A). These results not only reflected the transition of the three CD8⁺ subtypes but also revealed a shift in the cytotoxic and exhausted states of infiltrating T cells.

To investigate whether GOLP treatment resulted in a transition of CD8⁺ T-cell functional states, we calculated the exhaustion scores of iCCA patients treated with GOLP and found that CD8⁺ T cells in well response patients showed a more dramatic decrease in exhaustion scores than those in poor response patients during GOLP treatment (Fig. 6K). Furthermore, the successful transition from CD8 proliferating to CD8 GZMK⁺ led to a reduction in the exhausted scores of the well response patients, while the impeded conversion of CD8 proliferating or CD8 GZMB⁺ into CD8 GZMK⁺ led to an increase in exhausted scores in poor response patients (Fig. 6L and Supplementary Fig. 7A). A cell state transition from CD8 GZMB⁺ to CD8 GZMK⁺ has also been observed in coronavirus disease 2019 (COVID-19) patients[45]. In addition, we found that iCCA patients with higher CD8 GZMK⁺/CD8

proliferating ratios showed better prognosis in the large treatment-naive validation cohort (Fig. 6M). Similarly, we also found that patients with higher ratios of CD8 GZMK⁺/CD8 proliferating or CD8 GZMK⁺/CD8 GZMB⁺ tended to respond well to GOLP therapy (Fig. 6N). In addition, samples with high CD8 GZMK⁺/CD8 proliferating, CD8 GZMB⁺/CD8 proliferating and CD8 GZMK⁺/CD8 GZMB⁺ ratios showed positive Ti values (Fig. 6O). These results suggested that successful transition of CD8 proliferating and CD8 GZMB⁺ into CD8 GZMK⁺ might play a prominent role in achieving an effective response to GOLP therapy (Fig. 6P).

## Macro CD5L⁺ exacerbated the exhaustion of CD8 GZMB⁺ in patients with a poor response to GOLP

Considering the importance of the three subtypes of CD8⁺ T cells in GOLP therapy, we further explored the underlying mechanism of GOLP treatment in iCCA. Interestingly, RNA velocity analysis showed that patients with different responses to GOLP possessed obviously different CD8 GZMB⁺ clusters (Fig. 7A). During GOLP treatment, the CD8 GZMB⁺ cells in the patients with well response could transition into CD8 GZMK⁺ cells, whereas a substantial proportion of CD8 GZMB⁺ in the patients with poor response reverted into CD8 proliferating cells (Fig. 7A). In addition, CD8 GZMB⁺ in the well response group had a significantly higher proportion of clonal expansion than those in the poor response group after GOLP treatment (Fig. 7B). Simultaneously, the exhausted scores of CD8 GZMB⁺ were significantly decreased in the well response group after treatment but increased in the poor response group (Fig. 7C). By comparing the gene expression levels of CD8 GZMB⁺ between the PR and SD groups, we found that genes highly expressed in PR patients (*GZMH*, *TRBV9*, and *HLA-DQA2*) were enriched in the adaptive immune response, regulation of lymphocyte activation and antigen processing and presentation pathways, while genes enriched in the SD group (*STAT1*, *STLA4*, and *KLRB1*) were associated with cytokine signaling in the immune system and response to cytokines (Fig. 7D, E), suggesting that the exhausted state of CD8 GZMB⁺ in samples with poor responses to GOLP treatment might be affected by cytokine signaling. Additionally, single-cell regulatory network inference and clustering (SCENIC) analysis showed that CD8 GZMB⁺ in the poor response group had more TF regulon activities related to the STAT and IRF families (Supplementary Fig. 7E), which are key cytokines mediating T-cell differentiation[46]. These results suggested that the status of CD8 GZMB⁺ subtypes in samples with poor response to GOLP treatment was potentially induced by cytokine signaling.

Therefore, we next explored the cytokine signaling strengths activity among the identified cell types by CellChat analysis[47], and found that Macro CD5L⁺ showed the highest outgoing interaction strength, while CD8 GZMK⁺, CD8 GZMB⁺ and CD8 proliferating showed the highest incoming interaction strengths (Fig. 7F and Supplementary Fig. 7L). Notably, Macro CD5L⁺ showed the strongest interactions with CD8 GZMB⁺ through cytokine signaling pathways (Fig. 7G), indicating that Macro CD5L⁺ might play a role in the GOLP response through interaction with CD8 GZMB⁺. Additionally, the bulk RNA-seq data from

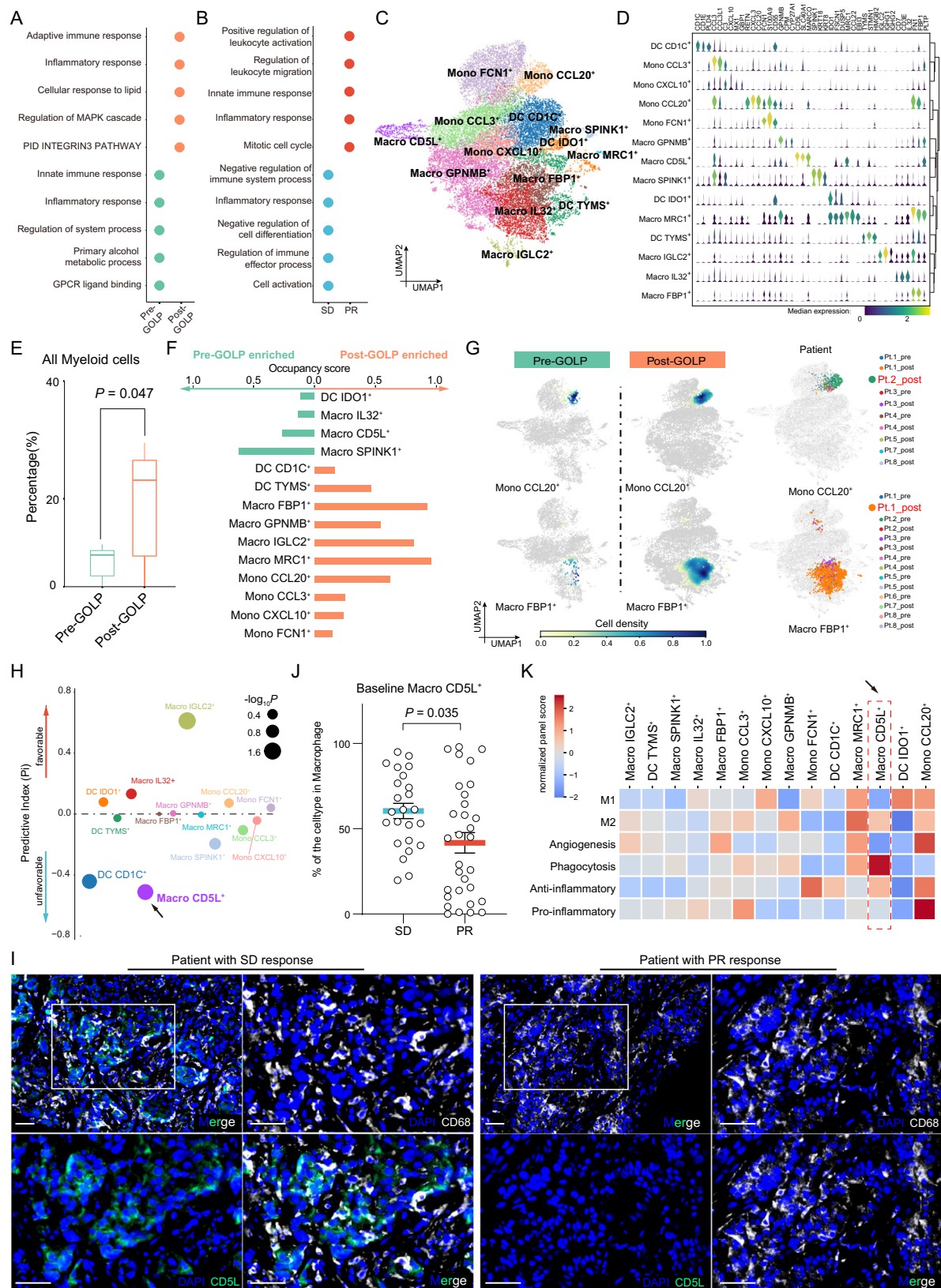

the FU-iCCA iCCA cohort showed that CD8 GZMB⁺ exhibited coordinated dynamics with Macro CD5L⁺ (Fig. 7H). The spatial colocalization between Macro CD5L⁺ and CD8 GZMB⁺ in the TME of iCCA was validated by mIHC analysis (Fig. 7I).

To validate the interaction between Macro CD5L⁺ and CD8 GZMB⁺ T cells directly, we sorted Macro CD5L⁺ and CD8⁺ T cells. Since CD5L is

a secreted protein, we choice FOLR2, a surface protein that specifically expressed on Macro CD5L⁺ across all the macrophages (Supplementary Fig. 7F), to sort Macro CD5L⁺. Interestingly, we found sorted CD14⁺ FOLR2⁺ macrophages showed completely different cell morphology with CD14⁺ FOLR2⁻ macrophages (Supplementary Fig. 7G), and we confirmed CD14⁺ FOLR2⁺ macrophages expressed high level of CD5L

**Fig. 4 | High proportion of Macro CD5L⁺ in tumors with poor response to GOLP.**
**A** Dot plot showing the top pathways significantly enriched in pre-GOLP and post-GOLP myeloid cells (N = 18 samples). A hypergeometric test was used. Benjamini–Hochberg (BH) adjusted *P* < 0.05 (two-sided) was used as a significance cutoff. **B** Dot plot showing the top pathways significantly enriched in myeloid cells of patients with SD or PR (N = 8 patients). A hypergeometric test was used. Benjamini–Hochberg (BH) adjusted *P* < 0.05 (two-sided) was used as a significance cutoff. **C** UMAP plot showing myeloid cells clustered into 14 subtypes (N = 20,825 cells). Macro, macrophage; DC, dendritic cell; Mono, monocyte. **D** Violin plot showing the marker gene expression levels of the 14 myeloid subtypes. The colors in the violin plot represent the median gene expression levels in the subtypes. **E** Pairwise comparison of overall myeloid cell proportions between the pre-GOLP and post-GOLP groups. Paired Wilcoxon test, two-sided. N = 8 biologically independent patients. The results are depicted in boxplots: center line indicates median, box represents first and third quantiles, and whiskers indicate maximum and

minimum values. **F** Bar plot showing the occupancy score of each myeloid subtype (N = 8 patients). **G** UMAP plot (N = 20,825 cells) showing the dynamics of Mono CCL20⁺ and Macro FBP1⁺ upon GOLP treatment (N = 8 patients). **H** Pi values of myeloid subtypes in iCCA patients treated with GOLP. N = 8 patients. The arrow points to Macro CD5L⁺ (*P* = 0.046), the baseline proportion of which was a strong predictor of unfavorable GOLP response (Wald test, two-sided). **I** Representative samples from patients with SD and PR (N = 2 samples) stained by IHC with anti-CD5L (green) and anti-CD68 (white) antibodies. Scale bar, 40 μm. **J** Dotplot showing the baseline Macro CD5L⁺ percentages of the SD and PR groups in three GOLP treatment cohorts (NCT03951597 and FDU-ZS-ICC-T). N = 55 biologically independent patients. Mann–Whitney test, two-sided, *P* = 0.035. The results are depicted with Mean ± SEM. **K** Heatmap showing previously defined macrophage signatures across all myeloid subtypes. Macro CD5L⁺ were highlighted by arrow as M2-polarized and anti-inflammatory macrophages with mostly phagocytosis-related functions. Source data are provided as a Source Data file.

(Supplementary Fig. 7H, I). After co-culture with CD5L⁺ macrophage (1:1) for 72 h, the percent of CD8 GZMB⁺ expressing CTLA4, KLRB1 and LAG3 were larger than that with CD5L⁻ macrophage significantly, but the percent of CD8 GZMB⁺ expressing PD1 showed no statistic difference (Fig. 7J). High expression levels of *CTLA4* and *LAG3* have been found to be associated with an exhausted state of T cells[48], while a high *KLRB1* expression level showed an innate-like and low cytotoxic phenotype of CD8⁺ T cells[21]. Additionally, we also found the supernatants of sorted Macro CD5L⁺ had significantly higher levels of CD5L protein compared to that of Macro CD5L⁻ cells (Supplementary Fig. 7J), and the addition of CD5L protein into the media of Macro CD5L⁻ co-cultured with CD8⁺ T cells significantly increased the expression level of CTLA4 of CD8 GZMB⁺ cells (Supplementary Fig. 7K). These results revealed the interaction between Macro CD5L⁺ and CD8 GZMB⁺ cells by CD5L protein could reshape the exhausted state of CD8 GZMB⁺ T cells in a PD1-independent manner.

Cell–cell interaction analysis by single-cell data further revealed the interactions between Macro CD5L⁺ and CD8 GZMB⁺ were mainly achieved by *CD5L* and *CXCL12* related pathways (Supplementary Fig. 7M; Supplementary Data 7; **Methods**). In particular, the CXCL12-associated CXCL signaling network showed the highest communication probabilities between Macro CD5L⁺ and CD8 GZMB⁺ (Supplementary Fig. 7L). Notably, three receptor genes (*CD5*, *CXCR3* and *CXCR4*) showed significantly higher co-expression values with the enriched TFs (*STAT1* and *IRF9*) in CD8 GZMB⁺ cells in samples with a poor response to GOLP treatment (Supplementary Fig. 7N), indicating potential activation of downstream TFs after Macro CD5L⁺ and CD8 GZMB⁺ communication. Notably, the downstream genes (*CTLA4*, *LAG3* and *KLRB1*) regulated by *STAT1* were also upregulated in CD8 GZMB⁺ in patients with SD (Supplementary Fig. 7O, P; Supplementary Data 7).

Together, these results suggested that Macro CD5L⁺ potentially activate the extracellular signals of CD8 GZMB⁺ and exacerbate the exhausted status of CD8 GZMB⁺ in patients with poor response to GOLP therapy.

**Anti-CTLA4 antibody reversed GOLP resistance in iCCA**
To confirm the role of Macro CD5L⁺ in GOLP therapy in vivo, we constructed mouse models of iCCA by liver orthotopic injection (Supplementary Fig. 8A) or subcutaneous injection (Supplementary Fig. 1F). We detected the presence of Macro CD5L⁺ in the tumor environment of liver orthotopic injection model, while hardly detected them in the subcutaneous model (Supplementary Fig. 8B, C), suggesting that Macro CD5L⁺ specifically existed in the liver environment. Similar to human iCCA, Macro CD5L⁺ was specifically co-expressed with membrane protein FOLR2 in the orthotopic injection model (Supplementary Fig. 8B, C), which can be used as a surface marker for specific sorting of Macro CD5L⁺. We sorted Macro CD5L⁺ (CD45⁺ F4/80⁺ FOLR2⁺) in the tumor environment of orthotopic injection model and injected the sorted Macro CD5L⁺ into the subcutaneous iCCA model to

directly explore its effect on GOLP therapy. Mouse iCCA showed poorer GOLP response in the Macro CD5L⁺ injection group than that in PBS or Macro CD5L⁻ injection control (Fig. 8A and Supplementary Fig. 8D, E), suggesting Macro CD5L⁺ led to the resistance of GOLP therapy in mouse. scRNA-seq showed that the subcutaneous iCCA environment after GOLP therapy can be divided into 8 main clusters: Granulocytes, T cells, B cells, Monocytes, Macrophages, Fibroblast, Endothelial cells and Malignant cells (Fig. 8B, Supplementary Fig. 8D). We further confirmed the absence of Macro CD5L⁺ in the control group in the single-cell transcriptome, while the Macro CD5L⁺ injection group had colonization of Macro CD5L⁺ in the tumor environment (Fig. 8B).

Among the six unsupervised clustering group 0–5 of CD8⁺ T cells, Clusters 0 and 1 showed the most similar gene expression profile to the human CD8 GZMB⁺, Cluster 2 was most similar to the human CD8 GZMK⁺, while Cluster 5 were most similar to the aforementioned human CD8 proliferating (Fig. 8C). This suggests that the three types of human CD8 T cell subsets are relatively conserved between human and mouse. Consistent with human CD8 GZMB⁺, the human-like CD8 GZMB⁺ clusters (Cluster0, 1) also had significantly higher *Ctla4*, *Tox* and *Tox2* expression in the tumor environment injected with Macro CD5L⁺, while the expression of *Pdcd1* (PD1) was not different between the two groups (Fig. 8D). However, unlike in the human tumor environment, the human-like CD8 GZMB⁺ clusters in mouse hardly expressed *Lag3* or *Klrb1* (Fig. 8D).

Considering that *Ctla4*, but not *Lag3* or *Klrb1* highly expressed both in mouse and human CD8 GZMB⁺ in the presence of Macro CD5L⁺ (Figs. 7J and 8D), we treated mouse with GOLP plus anti-CTLA4 antibody. As expected, anti-CTLA4 antibody can significantly reverse GOLP resistance due to Macro CD5L⁺ injection (Fig. 8E, F). Since *CTLA4* may also expressed in other cells than CD8 GZMB⁺, we re-analyzed scRNA data and found that *Ctla4* is also expressed in mouse FoxP3⁺ CD4 Treg cells (Supplementary Fig. 8G–I). However, different from CD8 GZMB⁺ (Fig. 8D), Macro CD5L⁺ did not increase expression of *Ctla4* in CD4 T cells or Tregs (Supplementary Fig. 8J), suggesting that the CTLA4-mediated reversal of GOLP resistance is independent of CD4 Tregs. To further confirm this conclusion, we depleted Tregs using an anti-CD25 antibody, and found that anti-CTLA4 treatment could still reverse Macro CD5L⁺-induced GOLP resistance (Supplementary Fig. 8K). Together, these results conclusively support the reversal effect of the treatment regimen was CD4 Treg independent, but CD8 GZMB⁺ dependent. mIHC analysis also showed higher CTLA4 expression in the poor responsive patients after GOLP therapy (Fig. 8G, H), suggesting CTLA4 is a potential target that can reverse GOLP resistance in future clinical practice.

## Discussion
Population screening and mechanistic exploration are important for the further optimization of combination therapy. Here, we generated single-cell profiles of tumor cells and the immune microenvironment

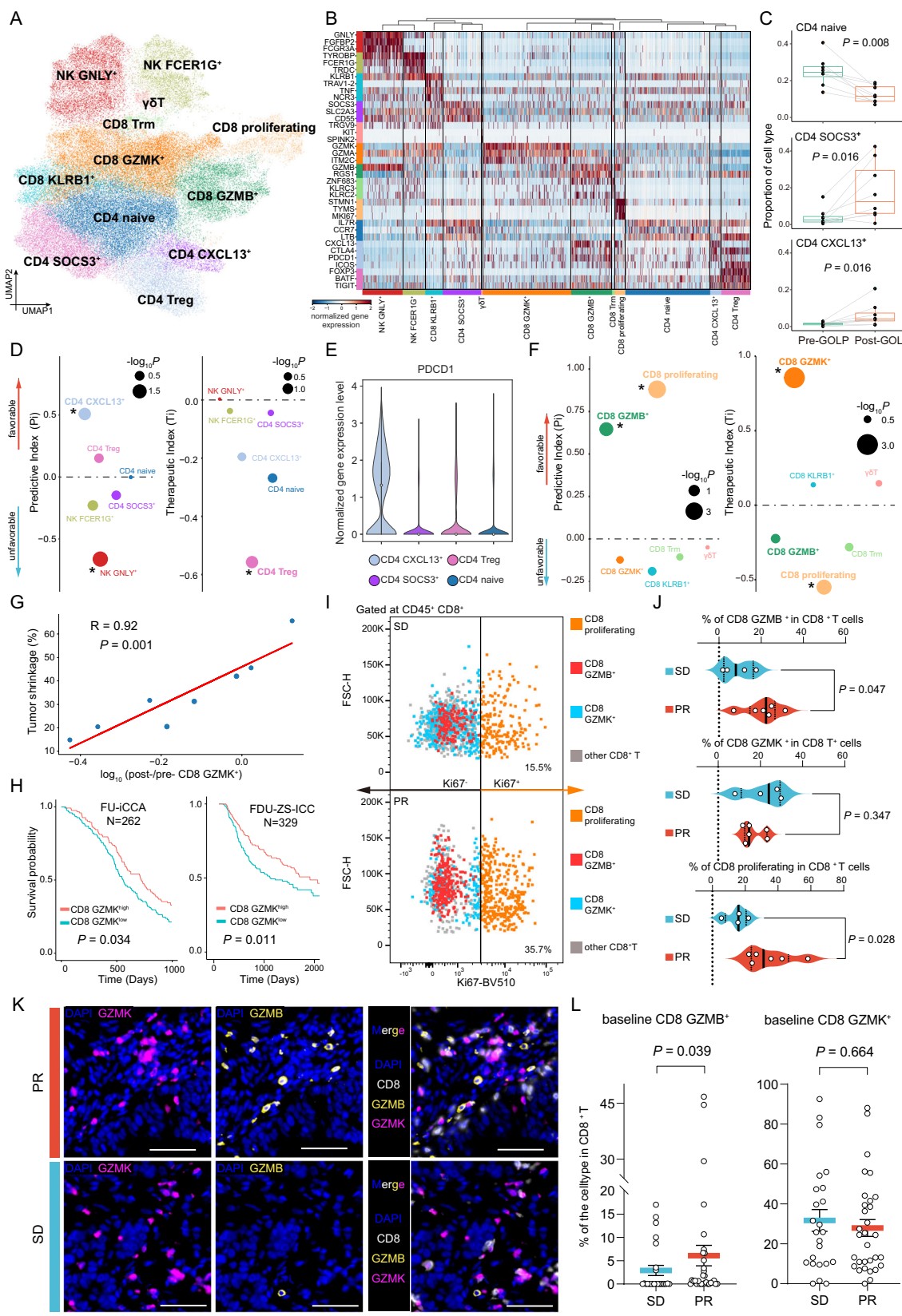

in paired samples from patients with iCCA treated with a high-ORR GOLP therapy. We found baseline CD8 GZMB[+] as a predictor of favorable response to GOLP therapy. We also revealed that the dynamic transition of CD8 proliferating and CD8 GZMB[+] into CD8 GZMK[+] was associated with distinct responses to GOLP treatment. More importantly, we identified that Macro CD5L[+] exacerbated the exhaustion of CD8 GZMB[+] by CTLA4 elevation, resulting in a poor response to GOLP therapy, and targeting CTLA4 reversed GOLP resistance in mouse iCCA. Our analyzes revealed key cell clusters linked to the efficacy and mechanism of GOLP therapy that can be further studied to optimize combination therapy in iCCA (Fig. 9 and Supplementary Fig. 9).

**Fig. 5 | High baseline levels of CD8 GZMB⁺ predicted a favorable response to GOLP treatment. A** UMAP plot (N = 49,053 cells) showing the 12 subtypes of T and NK cells (N = 8 patients). **B** Heatmap showing the normalized expression of marker genes in lymphoid subtypes. **C** Proportions of three CD4 subtypes between the pre-/post-GOLP. Paired Wilcoxon test, two-sided. N = 8 biologically independent patients. Depicted in boxplots: center line indicates median, box represents first and third quantiles, and whiskers indicate maximum and minimum values. **D** The Pi and Ti of CD4⁺ T and NK subtypes in iCCA treated with GOLP. NK GNLY⁺ showed a negative Pi (P = 0.014), CD4 CXCL13⁺ showed a positive Pi (P = 0.047), and CD4 Treg showed a negative Ti (P = 0.033). N = 8 paired samples. Wald test, two-sided. *: P < 0.05. **E** PDCD1 expression among CD4⁺ T-cell subtypes. N = 8 paired samples. Depicted in violin plots the dot indicates median. **F** Pi and Ti of CD8⁺ T-cell subtypes in iCCA tumors treated with GOLP. CD8 proliferating (P = 0.0006) and CD8 GZMB⁺ (P = 0.016) showed significantly positive Pi values. CD8 GZMK⁺ showed a positive Ti (P = 0.001), whereas CD8 proliferating (P = 0.0035) and CD8 GZMB⁺ (P = 0.23)

showed negative Ti values. N = 8 paired samples. Wald test, two-sided. *: P < 0.05. **G** Relationship between tumor shrinkage and the $\log_{10}$ normalized alteration ratio of the CD8 GZMK⁺ proportion after GOLP treatment (N = 8 paired samples, Wald test, two-sided, P = 0.001, R = 0.92). **H** Survival plot showing that high GZMK expression was associated with better prognosis in both the FU-iCCA validation cohort (N = 262, P = 0.034) and the FDU-ZS-ICC validation cohort (N = 329, P = 0.011). Log-rank test, two-sided. **I, J** Flow cytometry plot and analysis showing the percentages of the three CD8⁺ T-cell subtypes from patients with PR or SD pre-GOLP. t test, two-sided. N = 10 biologically independent patients. Depicted in violin plots: the center line indicates median; the dotted lines represent the first and third quantiles. **K, L** Representative mIHC and statistics of CD8⁺ T-cell subtypes from patients with PR or SD pre-GOLP. Scale bar, 50 μm. N = 55 biologically independent patients, Mann–Whitney test, two-sided. Depicted in dotplots with Mean ± SEM. Source data are provided as a Source Data file.

Our results showed that GOLP could achieve substantial decreases in tumor size in iCCA patients as well as reshape the iCCA TME. The inflammatory response and adaptive immune response were activated against residual tumor cells after GOLP treatment, which could be attributed to the exposure of tumor antigens and immunogens by gemcitabine[49]. In addition, the alteration of immune cells also indicated a transition into an immune active TME with GOLP therapy, which was characterized by a decrease in DC IDO1⁺ and CD4 naive, an increase in CD4 CXCL13⁺, and CD8 T-cell TCR expansion. The IDO1 gene has been reported to be an immune checkpoint molecule that inhibits DCs and effector T cells and activates Treg cells[32,33,50], suggesting that the decrease in DC IDO1⁺ after GOLP therapy released the brake of the suppressive TME. In contrast, CD4 CXCL13⁺ have recently been identified as a favorable cluster in response to anti-PD1 therapy in TNBC[18], supporting the increase in CD4 CXCL13⁺ cells after GOLP treatment in patients with a favorable response. In addition, the activation of CD8⁺ T-cell subtypes revealed by TCR expansion was consistent with a previous report of anti-PD1 therapy[16]. The increased myeloid infiltration and adaptive immune response in the TME after GOLP treatment was not only caused by the anti-PD1 antibody. Lenvatinib can also modulate the immune status in the TME by reducing Tregs and macrophages and increasing CD8⁺ T-cell infiltration[51,52]. Moreover, oxaliplatin and gemcitabine can also activate macrophages and reprogram tumor-associated macrophages (TAMs) toward an immunostimulatory phenotype[53,54]. Therefore, the findings suggest that the unique TME generated after GOLP therapy was actually caused by a combination of the targeted and conventional chemotherapies in addition to the anti-PD1 antibody.

Another finding of our study is the identification of tumor and immune features associated with different responses to GOLP therapy. We identified the C1 meta-cluster of tumor cells, which featured the function of antigen presentation and processing, as a predictor of favorable GOLP response. In tumor cells, we also found that high baseline expression of specific genes, including *PCNA*, *FGF11*, *FGF18*, *CD274* (*PD-L1*) and *HLA-A*, was associated with a favorable response. These genes were found to be enriched in PD1 signaling, vasculature development and signaling by KIT pathways. *PCNA* is found to be upregulated in proliferative tumor cells and a marker of sensitivity to chemotherapy[55,56]. As direct targets of lenvatinib[57], *FGF* and *KIT*-related pathways are related to tumor growth, angiogenesis, and metastasis. PD1/PD-L1 activation in tumors can be blocked by anti-PD1 antibodies. Together, the findings indicate that in GOLP treatment, the chemotherapy can rectify aberrant cell proliferation, lenvatinib can rectify abnormal angiogenesis and KIT signaling, and the anti-PD1 antibody can reverse the suppressive immune status, thus benefiting iCCA patients with high levels of these molecular features. We identified favorable (CD4 CXCL13⁺, CD8 GZMB⁺ and CD8 proliferating) and unfavorable (CD4 Treg, NK GNLY⁺ and Macro CD5L⁺) immune cell subtypes predicting response to GOLP treatment. CD4 CXCL13⁺ were

also recently reported to be a cluster predicting a favorable response to anti-PD1 therapy in TNBC[18]. Mechanistically, CD4 CXCL13⁺ can also recognize tumor neoantigens and activate macrophages, CD8⁺ T cells, and memory B cells[58], supporting their favorable role in the antitumor immune response. GZMB has been identified as a cytotoxic T lymphocyte (CTL) marker that induces tumor apoptosis[59]. Additionally, GZMB⁺ CD8 T cells were reported to be an activated CD8 T cluster in breast cancer[60]. In combination with the clonal expansion observed in the CD8 GZMB⁺ subtype upon GOLP therapy, these results support CD8 GZMB⁺ as a predive factor and therapeutic target worthy of further exploration in iCCA.

We also uncovered a potential mechanism for the distinct responses to the GOLP regimen in iCCA. The successful transition of CD8 proliferating into CD8 GZMK⁺ with CD8 GZMB⁺ as an intermediate was associated with a good response to GOLP therapy, while impediment of this process was associated with a poor response. Lack of this transition in SD patients was a result of low clonal expansion and high exhaustion of CD8 GZMB⁺, and the exhausted status of CD8 GZMB⁺ was caused by their communication with Macro CD5L⁺. Evidence has shown that targeted and conventional chemotherapy can modulate macrophage status[24]. Here, our results suggest that Macro CD5L⁺ increase the exhaustion of CD8 GZMB⁺ through the CD5L and CXCL12/STAT pathways. Macro CD5L⁺ have been reported to promote macrophage M2 polarization[37], which could explain why patients with high baseline Macro CD5L⁺ showed a poor response to GOLP. In addition, activation of the CXCL12/CXCR4 axis can impair CD8⁺ T cells, and targeting this axis facilitated anti-PD1 immunotherapy in pancreatic cancer[61–64]. Previous reports also showed that activation of CXCL12/CXCR4 axis can trigger downstream biological effects mediated by STAT activation[65], including resistance of T cells to immunotherapy[66] and decreased infiltration of effector T cells into the TME after anti-PD1 therapy[67]. In particular, STAT signaling can enhance PD1 expression in T cells following cytokine stimulation[68], indicating that these pathways could influence the exhaustion status of T cells. Thus, our results show that the crosstalk between Macro CD5L⁺ and CD8 GZMB⁺ induced a poor response to the GOLP regimen in patients with iCCA potentially through the CXCL12-CXCR4/STAT pathway.

The exhausted status of CD8 GZMB⁺, characterized by high expression of *CTLA4*, *LAG3*, and *KLRB1* in addition to *PDCD1*, was highlighted in poor response to GOLP. We identified Macro CD5L⁺ led to the increased of *CTLA4* in CD8 GZMB⁺, and targeting *CTLA4* can reverse GOLP resistance in mouse models. The increase of CTLA4 by Macro CD5L⁺-CD8 GZMB⁺ crosstalk is a PD1-independent exhaustion of CD8 T cells, thus PD1 antibody in GOLP therapy cannot exert its therapeutic effects in this phenomenon, indicating that targeting Macro CD5L⁺-CD8 GZMB⁺ crosstalk or a multi-immune-checkpoint regimen (especially anti-CTLA4 antibody) that prevents CD8 GZMB⁺ exhaustion could potentially benefit patients who are insensitive to GOLP.

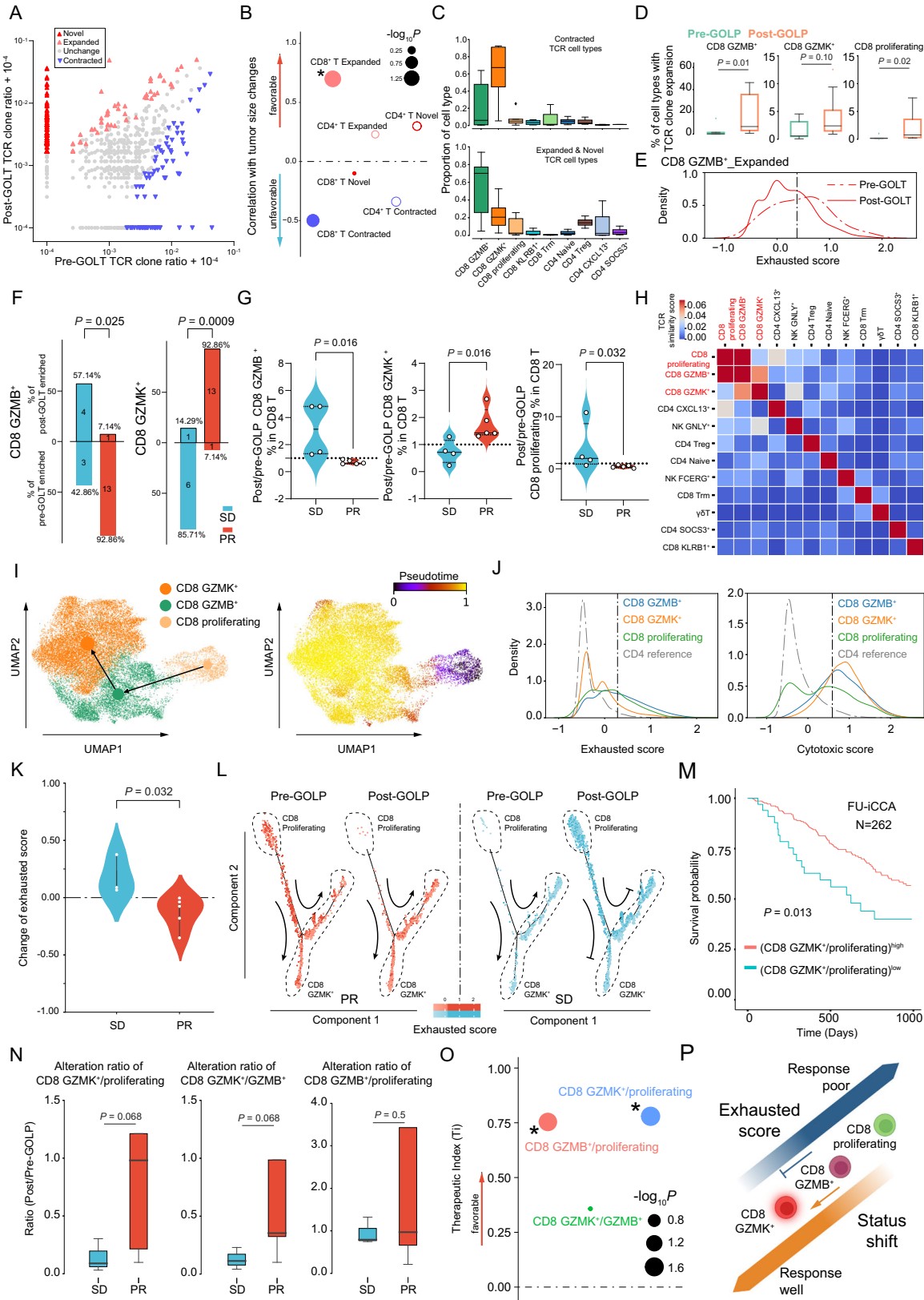

The limits of single-drug therapy in iCCA have been well recognized, which hinders the possibility of designing clinical trials with only one constituent regimen of GOLP for more stringent comparison. The sample size in this study was limited by the difficulty in obtaining paired samples with GOLP therapy for scRNA-seq analysis. We have

substantially circumvented the problem by validating the main conclusions in large cohorts as well as in vitro and in vivo experiments. Another limitation is the efficacy of GOLP therapy from phase II trial needs to further validate in phase III trial. We have begun a multicenter, double-blinded, randomized, phase III study to confirm the high

**Fig. 6 | TCR clonotype expansion and influence of CD8 proliferating to CD8 GZMB⁺ to CD8 GZMK⁺ transition on GOLP response. A** TCR clonal types classified by the change upon GOLP therapy (N = 25,893 types, N = 8 patients, Fisher's exact test, two-sided). See exact *P* in Supplementary Data 8. **B** Correlation between tumor change and clonotype status in different T-cell types. N = 8 paired samples, Wald test, two-sided. *: *P* < 0.05. See exact *P* in Source Data. **C, D** Distribution of contracted/expanded clonotypes among the T-cell subtypes and proportions of CD8⁺T subtypes with clonal expansion between the pre- and post-GOLP groups (Wilcoxon rank-sum test, two-sided. N = 8 independent patients). **E** Exhausted scores of CD8 GZMB⁺ cells with clonal expansion pre- and post-GOLP. Gray dashed line cutoff: exhausted score calculated from CD4 SOCS3⁺ and CD4 naive cells (N = 8 paired samples, 99% confidence level, one-tail test). **F** Distribution of CD8 GZMB⁺ and CD8 GZMK⁺ in the pre- or post-GOLP groups between patients with PR and SD (N = 21) derived from mIHC analysis in GOLP treatment cohorts. Fisher's exact test, two-sided. **G** Flow cytometry analysis showing the alteration ratio of CD8⁺T-cell subtypes from patients with PR or SD post-/pre-GOLP (N = 9). Mann–Whitney U test,

two-sided. **H** TCR clonal similarity across T/NK subtypes. **I** RNA velocity analysis showing the transition among the three CD8⁺T cells (N = 19,128 cells). **J** Distribution of exhausted and cytotoxic scores of the three CD8⁺T-cell subtypes pre-/post-GOLP treatment (Reference: CD4⁺T-cell). **K** Exhausted score of CD8⁺ T cells between PR and SD patients upon GOLP treatment. Mann–Whitney U test, two-sided, N = 8 independent patients. **L** Change in exhausted score along the trajectory in two representative patients pre-/post-GOLP. **M** Overall survival with different CD8 GZMK⁺/CD8 proliferating ratio (N = 262, log-rank test, two-sided). **N** Alterations in ratios of three CD8⁺ T cells in the pre- or post-GOLP groups between patients with PR and SD. Mann–Whitney U test, two-sided, N = 8 independent patients. **O** Ti of changes in the CD8 GZMK⁺/CD8 proliferating (*P* = 0.023), CD8 GZMK⁺/CD8 GZMB⁺ (*P* = 0.385), and CD8 GZMB⁺/CD8 proliferating ratios (*P* = 0.031). N = 8 paired samples, Wald test, two-sided. *: *P* < 0.05. **P** The status and phenotypic shift in CD8⁺ T cells. For all boxplots: center line indicates median, box represents first and third quantiles, and whiskers indicate maximum and minimum values. Source data are provided as a Source Data file. Created with BioRender.com.

efficacy of this combination therapy in patients with advanced iCCA, with NMPA approval (No. 2021LP01825) and Clinicaltrials.gov registration (No. NCT05342194).

In this work, we elucidate the dynamic atlas during the treatment of GOLP in patients with iCCA. Our study identifies baseline CD8 GZMB⁺ level as a predictor of favorable response to GOLP treatment. The transition of CD8 T-cell subtypes mediated by the CD8 GZMB⁺-Macro CD5L⁺ interaction is the key mechanism underlying the different responses to the GOLP regimen in iCCA, and targeting CTLA4 is potential to reverse GOLP resistance in the future clinical practice, thus potentially enabling the optimization of combination therapy.

## Methods

All experimental protocols described in this study was approved by the Zhongshan Hospital Research Ethics Committee and complied with all ethical regulations.

### Key resources table

Detail of the key regent or resource used in this paper can be found in Supplementary Data 9.

### Human subject, GOLP treatment and response evaluation

A total of 55 patients with GOLP treatment in the two cohorts were enrolled in this study (27 patients in NCT03951597, 28 patients in FDU-ZS-iCCA-T). Another 591 patients in the two GOLP treatment-naive cohorts (262 patients in FU-iCCA[69] and 329 patients in FDU-ZS-iCCA) were also enrolled into this study. All the patients were diagnosed with iCCA pathologically at Zhongshan Hospital, Fudan University. Detailed clinical information of the patients was showed in Supplementary Data 1. Written informed consent for clinical information (including age, gender and other clinical information) collection and publication, tissue collection were obtained from all the patients. The collection of human samples and clinical information was approved by the Zhongshan Hospital Research Ethics Committee (B2020-157(2), B2022-480R).

All the patients under GOLP treatment were treated with the following therapeutic strategy, lenvatinib (8 mg/d) daily, PD1 antibody (Toripalimab 240 mg) at Day1, oxaliplatin 85 mg/m² + gemcitabine 1 g/m² at Day1 and gemcitabine 1 g/m² at Day 8 for every 3-week treatment cycle. Paired pre-treated needle-biopsies and post-treated surgical samples were obtained before treatment (within 2 weeks) and after treatment. The ORRs of treatment were evaluated by medical imaging examinations at the baseline and after 3 cycles of treatment according to RECIST1.1[70]. Sex/gender analysis was not performed in this study since our previous study found subgroups stratified by sex presented similar ORRs in iCCA patients under GOLP therapy[11].

### Tissue dissociation

Lived tumor samples before and after GOLP therapy were collected in the RPMI 1640 Medium (Gibco), and enzymatically digested with gentle MACS Tumor Dissociation Kit, human (Miltenyi Biotec #130-095-929) for 30 min on a rotor at 37°C according to the manufacturer's protocol. The dissociated cells were next passed through a 40-μm cell-strainer (Corning) in the RPMI 1640 medium with 10% FBS (Gibco) until uniform cell suspensions were obtained. Subsequently, the suspended cells were passed through cell strainers and centrifuged at 300 × *g* for 5 min. The red blood cells were removed by Red Blood Cell Lysis Solution (Solarbio). After washing twice with 1× PBS (Sangon Biotech), the cell pellets were re-suspended in sorting buffer (PBS supplemented with 2% FBS).

### Single-cell RNA-seq library and TCR-seq library preparation and sequencing

Single-cell RNA-seq libraries were prepared using the Chromium Next GEM Single Cell 5' Kit v2 from 10x Genomics, following the manufacturer's instructions. In brief, single cells were washed once with PBS containing 0.04% bovine serum albumin (BSA) and resuspended in PBS containing 0.04% BSA to a final concentration of 500 ‒ 1200 cells/mL as determined by Rigel S2 cell counter (Countstar). 10,000 cells were captured in droplets to generate nanoliter-scale Gel bead in EMulsion (GEMs). GEMs were then reverse transcribed in a C1000 Touch Thermal Cycler (Bio-Rad) programmed at 53°C for 45 min, 85°C for 5 min, and held at 4°C.

After reverse transcription and cell barcoding, emulsions were broken and cDNA was isolated and purified with DynaBeads and SPRIselect reagent (Beckman Coulter), followed by PCR amplification. Amplified cDNA was then used for 5' gene expression library construction and TCR V(D)J targeted enrichment with the Chromium Single Cell Human TCR Amplification Kit (10x Genomics), followed by V(D)J library construction. For RNA-seq library construction, amplified cDNA was fragmented and end-repaired, size-selected, PCR-amplified with sample indexing primers, and double-sided size-selected. For TCR-seq library construction, TCR transcripts were enriched from amplified cDNA by multiplex PCR. Subsequently, enriched PCR product was fragmented and end-repaired, double-sided size-selected, PCR-amplified with sample-indexing primers, and size-selected. Libraries prepared according to the manufacturer's user guide were then purified and profiled for quality assessment. Single-cell RNA and TCR V(D)J libraries were sequenced by an Illumina Novaseq6000 sequencer with 150 bp paired-end reads.

### Cell lines and culture

Cell line mIC-22 was constructed by our group. Plasmids (1 μg Sleeping Beauty, 20 μg NICD and 4 μg AKT) were delivered to a C57/BL6J mouse

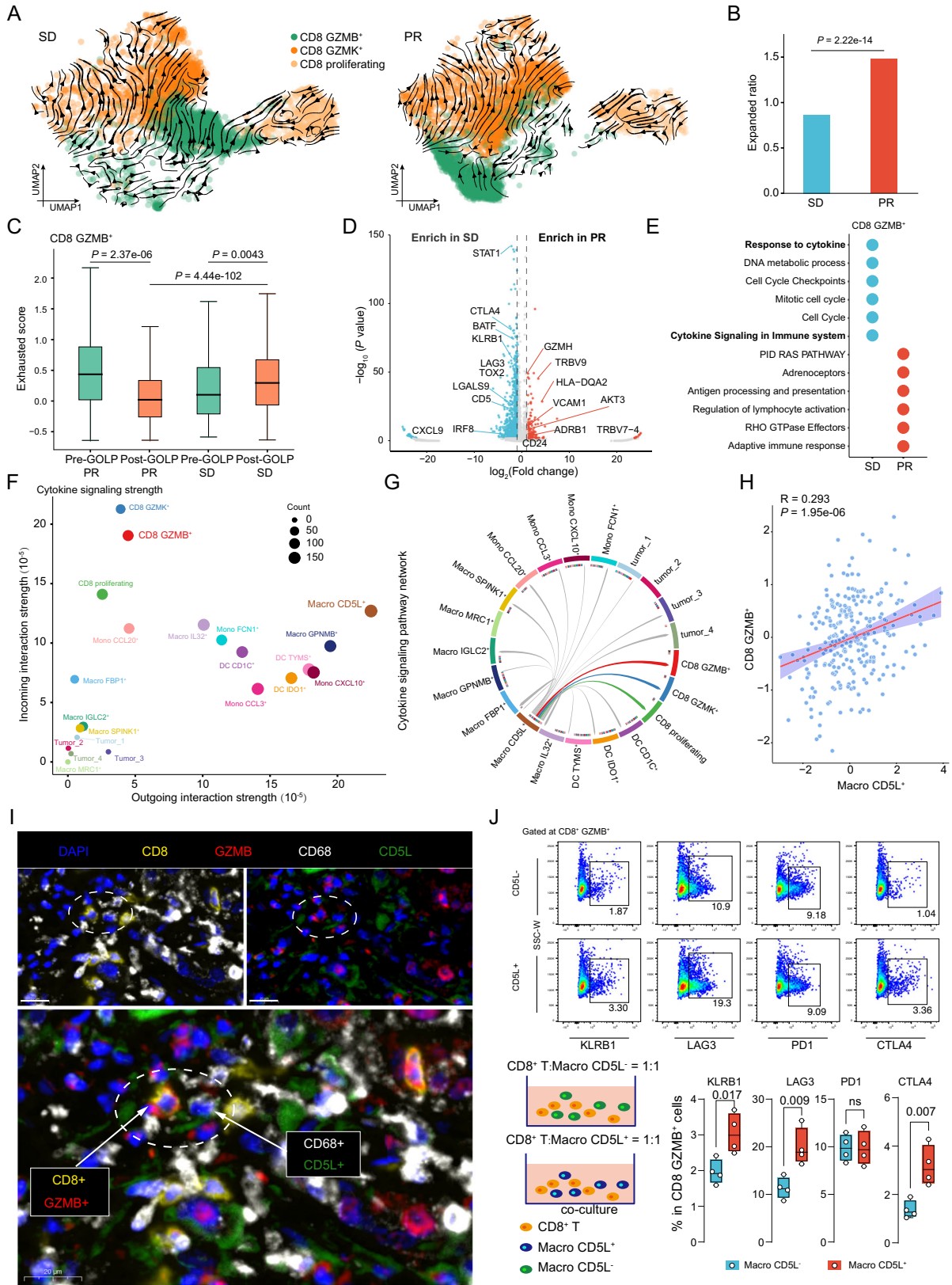

(male, 10 weeks old) by hydrodynamic tail vein injection. The mouse was then sacrificed and the tumor in liver was collected after 5 weeks. The tumor was then cut into small pieces, and cultured in vitro with 10% FBS 1640 medium for the following 6 months and the remaining tumor cells were collected. The mouse cell line AY-LTC2 was provided by Prof. Liu as a gift. Their tumorigenicity and origin of iCCA were verified by subcutaneous tumor formation and CK19 staining. All the cell lines were free of mycoplasma. The cell lines mIC-22 and AY-LTC2 were cultured in RPMI-1640 with 10% fetal bovine serum. All the cells were cultured at 37 °C in a 5% $CO_2$ incubator with 95% humidity.

**Fig. 7 | Macro CD5L⁺ exacerbated the exhaustion of CD8 GZMB⁺ in patients with a poor response to GOLP. A** The transition trajectory of CD8 proliferating, CD8 GZMB⁺ and CD8 GZMK⁺ inferred by RNA velocity analysis of samples from patients with SD and PR (N = 19,128 cells). **B** Proportion of CD8 GZMB⁺ with clonal expansion in SD and PR samples. Fisher's exact test, two-sided. **C** Distribution of exhausted scores of CD8 GZMB⁺ in PR and SD samples taken before and after GOLP treatment. Mann–Whitney U test, two-sided, 5631 cells from N = 8 biologically independent patients. **D** Differentially expressed genes of CD8 GZMB⁺ between the PR and SD group. Benjamini–Hochberg adjusted $P < 0.05$ was used as a significance cutoff. t test, two-sided. **E** Top enriched pathways in CD8 GZMB⁺ in the PR and SD groups. Benjamini–Hochberg adjusted $P < 0.05$ was used as a significance cutoff. Hypergeometric test, two-sided. **F** Dot plot showing the cytokine signaling strength of CD8 GZMB⁺, CD8 GZMK⁺, CD8 proliferating and the other 14 myeloid subtypes and tumor clusters (C1-C4), N = 8 paired samples. **G** CellChat analysis showing the network of cytokine signaling pathways in CD8 GZMB⁺, CD8 GZMK⁺, CD8

proliferating and the other 14 myeloid subtypes and tumor meta-clusters (C1-C4). The signal level from Macro CD5L⁺ was highest among the signals to CD8 GZMB⁺, and the signal level to CD8 GZMB was highest among the signals from Macro CD5L⁺. **H** Scatter plot showing correlations between CD8 GZMB⁺ and Macro CD5L⁺ in the FU-iCCA cohort (N = 262) based on signature gene expression. The regression line ± the 95% confidence interval was displayed, Wald test, two-sided. **I** A representative sample (N = 1 patient) showing crosstalk between CD8 GZMB⁺ and Macro CD5L⁺, which were stained by mIHC with anti-CD8 (yellow), anti-GZMB (red), anti-CD68 (white) and anti-CD5L (green) antibodies. Scale bar, 20 µm. **J** Co-culture strategy of CD8⁺ T and Macro CD5L⁺ or the control (Left). The expression of CTLA4, KLRB1 and LAG3 in CD8 GZMB⁺ after co-culture (Right, t test, two-sided, N = 4 biologically independent experiments for each group). For all boxplots: center line indicates median, box represents first and third quantiles, and whiskers indicate maximum and minimum values. Source data are provided as a Source Data file.

## Animal models

A total of $5 \times 10^6$ mIC-22 or AY-LTC2 were subcutaneously injected per C57/BL6J mouse (male, 6–7 weeks old). Treatment for the mice bearing tumor was performed 14 days after the subcutaneous injection. Anti-PD1 antibody were injected 200 µg/mouse intraperitoneally, twice a week; Oxaliplatin (5 mg/kg) and Gemcitabine (10 mg/kg) were injected intraperitoneally once a week; Lenvatinib (100 µg/mouse) was delivered five days a week by intragastric administration. To deplete Treg cells, anti-mouse CD25 (500 µg/mouse) was injected intraperitoneally three times a week, for 3 weeks. Anti-CTLA4 antibody (200 µg/mouse) were injected intraperitoneally, twice a week, for 3 weeks. All the animal experiments in this study were approved by the Zhongshan Hospital Research Ethics Committee. Maximal tumor diameter <20 mm was permitted by Zhongshan Hospital Research Ethics Committee and we confirmed that the maximal tumor size was not exceeded. Only male mice were used in this study to avoid potential risk of tissue rejection since mIC-22 cell line was constructed by a male C57/BL6J mouse.

## Data processing of scRNA-seq libraries and TCR-seq libraries

The scRNA-seq reads were aligned to the GRCh38 reference genome and quantified using cellranger count (10× Genomics, v6.0.2). Filtered and qualified feature matrix that only contained barcodes with unique molecular identifier (UMI) counts that passed the threshold for cell detection were used for further analysis. TCR reads were aligned to the GRCh38 reference genome and consensus TCR annotation was performed using cellranger vdj (10× Genomics, v6.0.2).

## Preprocessing and clustering iCCA single-cells

Further analyzes were performed using Scanpy (v1.8.2). Cells with less than 500 genes detected or greater than 8% mitochondrial RNA content were excluded from this analysis. Cells with n_feature greater than 6000 were also excluded. Possible cell doublets were predicted and excluded using package Scrublet[71]. Genes detected in fewer than 3 cells were excluded and the remaining 28,298 genes were used for the following analysis. After filtering, the median number of genes detected in cells is 1283.

For further clustering of all cell types in iCCA, qualified raw count data was log-normalized per cell with a size factor of 10,000. The top 3000 genes with the highest variations among all the cells were selected as "highly variable genes". Using these highly variable genes, a subset of the expression matrix was extracted and scaled. We then applied principal component analysis (PCA) to obtain meaningful features out of the scaled data. By plotting the elbow point of the PCA variance ratio distributions, sixteen PCs were selected to compute the neighboring cluster graph. Uniform Manifold Approximation Projection (UMAP) with leiden algorithm was applied for further dimensional reduction (resolution=0.8). In total, 34 clusters were obtained and visualized using UMAP. Next, we calculated differentially expressed

genes in each cluster using method 'wilcoxon rank sum-test'. After ranking, the top differentially expressed genes in each cluster were defined as marker genes and used for the following cell type annotations. Despite stringent filtering standards, we still identified a cluster with high mitochondrial gene expression and a cluster with high ribosomal gene expression (clusters 24 and 27), which were excluded from further analyzes. The remaining 32 (0-31) clusters were reordered and underwent SCSA automated cell-type annotation to aid less subjective annotation, while manual annotation based on prior knowledge in immunology and previous reports was used to improve the auto-annotation (Zhang et al. 2020; Cao et al. 2020).

A similar computational workflow was applied to sub-clustering and annotation of myeloid and lymphoid cell populations. In detail, we set Resolution=0.5 for both myeloid cells and lymphoid cells in UMAP re-clustering. The subclusters were initially assigned to major subtypes by classic gene markers in immunology[35,72]. Within these classic major subtypes, we dissociated and defined smaller clusters by their gene signatures. We assigned biologically meaningful functions to a few subtypes, e.g. CD4⁺ Treg according to previous reports, and other clusters were named by their top differentially expressed gene markers[72–74].

To compare the major cell type alterations upon GOLP treatment, we calculated enrichment scores to demonstrate the enrichment of each cell type in pre- or post-GOLP treated samples. The enrichment score was defined as Eq. 1,

$$\text{enrichment score} = \left| 1 - \left( \frac{pre}{Npre} \right) \Big/ \left( \frac{pre + post}{Npre + Npost} \right) \right| \qquad (1)$$

where pre and post represent cell-type specific cell numbers in pre- and post-GOLP treated samples; Npre and Npost represent total cell numbers in pre- and post-GOLP group, respectively.

## Definition of predictive index and therapeutic index

To circumvent the limitation of sample sizes, we applied predictive index (Pi) and therapeutic index (Ti) to leverage the single-cell and tumor size changes in a continuous axis. The Pi was designed to determine whether a baseline cellular proportion was associated with tumor size changes. The tumor size changes were calculated by the relative changes of tumor sizes between post- and pre-GOLP samples. For each baseline cell type, a linear regression model lm (y ~ x) was built. Using the predictive model information, Pi was defined as Eq. 2,

$$Pi = \frac{-slope}{|slope|} \times R^2 \qquad (2)$$

A positive Pi represents that a higher baseline level of the corresponding cluster was related with a better response to the GOLP.

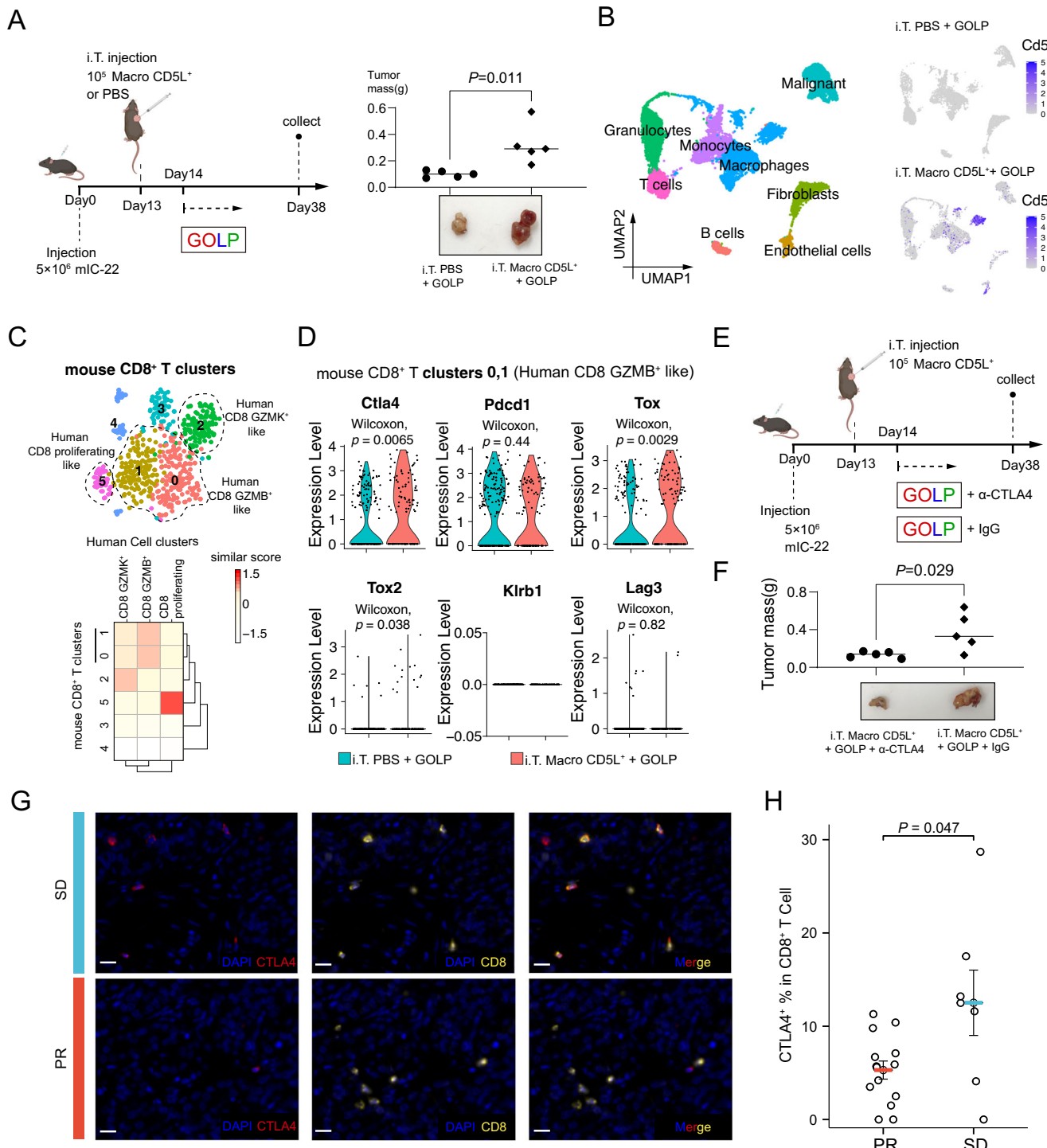

**Fig. 8 | Anti-CTLA4 antibody reversed GOLP resistance in iCCA. A** Workflow of mouse iCCA treatment and the analysis of tumor mass between Macro CD5L[+] injection group and the control group (N = 5 biologically independent animals for each group, t test, two-sided). Created with BioRender.com. **B** scRNA-seq confirming the absence of Macro CD5L[+] in the control group, while the Macro CD5L[+] injection group had colonization of Macro CD5L[+] in the tumor environment (N = 32,285 cells). **C** scRNA-seq showing the six unsupervised clustering group 0-5 of mouse CD8[+]T cells (Top, N = 572 cells); Similar score between the six unsupervised clustering group in mouse CD8[+]T cells and the three key human CD8[+]T cells (Bottom). **D** The gene expression of *Ctla4, Pdcd1, Tox, Tox2, Klrb1* and *Lag3* in the human-like CD8 GZMB[+] clusters (mouse CD8[+]T Cluster0, 1) by scRNA-

seq (N = 305 cells, Wilcoxon test, two-sided). **E, F** Workflow of mouse iCCA treatment and the analysis of tumor mass between the anti-CTLA4 group and the control group (N = 5 biologically independent animals for each group, t test, two-sided). Created with BioRender.com. **G** Representative samples (N = 2 samples) from patients with PR or SD after GOLP stained by mIHC with anti-CD8 (yellow), anti-CTLA4 (red) antibodies. Scale bar, 20 μm. **H** Quantitative comparison of CTLA4 expression in CD8[+]T cells between the PR and SD groups using mIHC results in the GOLP treatment cohorts with post-GOLP surgery (NCT03951597 and FDU-ZS-ICC-T) (total of 55 patients, including 21 with post-GOLP surgery; Mann–Whitney U test, two-sided). The results are depicted with Mean ± SEM. Source data are provided as a Source Data file.

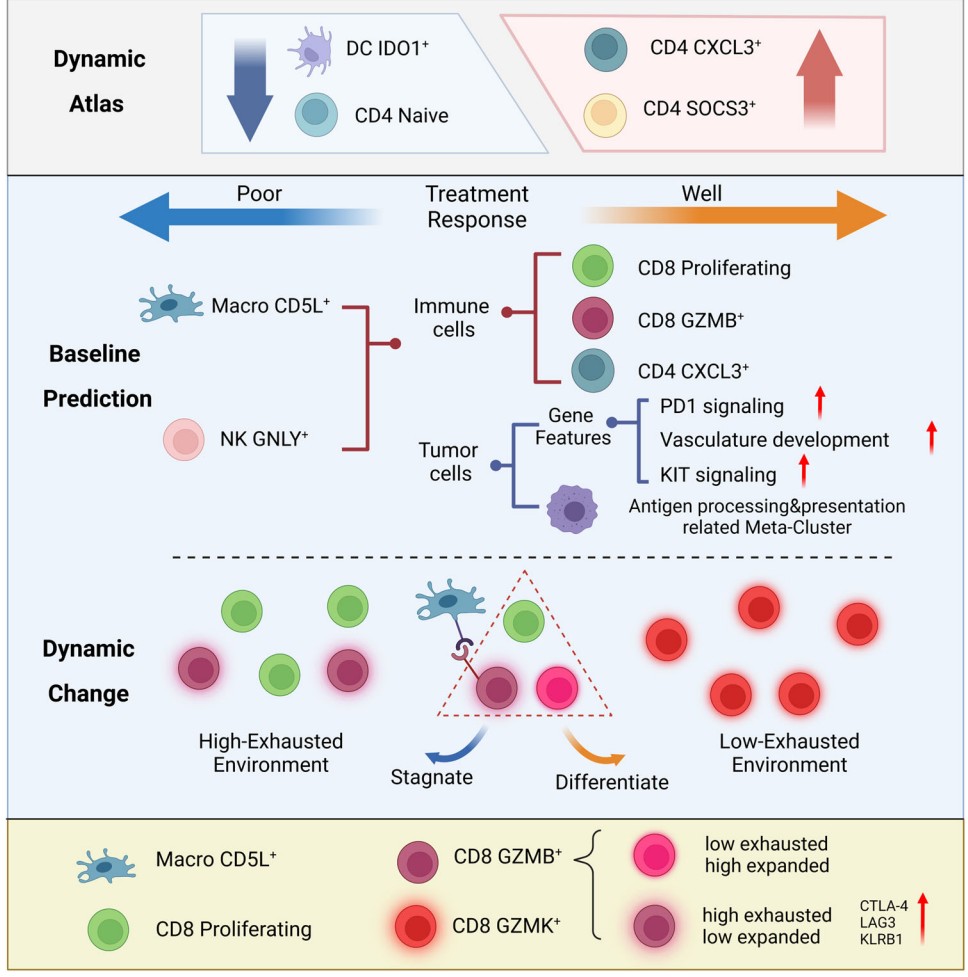

**Fig. 9 | Summary of the findings of this study.** This study highlights: (1) High proportions of CD8 GZMB⁺ and CD8 proliferating cells predicted good responses to GOLP treatment in iCCA; (2) Transition of CD8 proliferating to CD8 GZMK⁺ facilitated a favorable GOLP response in iCCA; (3) Macro CD5L⁺ and CD8 GZMB⁺ crosstalk impaired GOLP response by CTLA4 up-regulation. Created with BioRender.com.

The Ti was calculated to determine whether the dynamic change of cellular proportions was related with the tumor size changes. The cellular proportion dynamics were calculated by the relative changes between post- and pre-GOLP samples. To increase the stability of the liner model, the log10 value of dynamic changes was used to fit the lm (y ~ x) model. Similar to Pi definition, Ti was defined as Eq. 3,

$$Ti = \frac{-slope}{|slope|} \times R^2 \qquad (3)$$

by using therapeutic model information. A positive Ti represents a higher level of increase in cellular proportion for the corresponding cell cluster was associated with better clinical response, while a negative Ti represents that a higher level of increase in proportion for the cell cluster is associated with a worse clinical response.

### Gene set variation analysis and enrichment analysis
Functional enrichment analyzes were carried out to evaluate activation of hallmark pathways and metabolic pathways, which were described in the molecular signature database and the curated dataset, respectively[75,76]. Then, we applied gene set variation analysis (GSVA v1.26.0) to assign pathway activity estimates to individual cells[77]. Normalized GSVA scores were obtained for further heatmap visualization. In this study, DEGs under different conditions between pre- and

post-GOLP group or between SD and PR groups were obtained and enriched using Metascape[78].

### Single-cell CNV detection
Single-cell CNVs were detected using InferCNV (v1.6.0) with raw count matrix of single-cell RNA-seq expression as input[27]. Using Endothelial and fibroblast counts as reference, 'infercnv::run' was used to infer the CNV in tumor cells (Epithelial cells) with cutoff = 0.1.

### Sample level occupancy of cell cluster (SLOCC) calculation
For each cluster, we calculated the sample level occupancy, defined as Eq. 4,

$$SLOCC = \frac{\max(pro)}{\sum_1^n pro} \qquad (4)$$

where 'pro' represented cell counts of a cluster in a sample[79]. The value of SLOCC reflected the sample contribution for a particular cluster.

### Non-negative matrix factorization of tumor cells
To identity gene expression signatures representing the intra- and inner- tumor transcriptional heterogeneity of tumor cells, consensus non-negative matrix factorization (cNMF)[29] was applied. All tumor cells were decomposed into four clusters (k = 4). The k value was determined by considering the trade-off between stability and error[80]. Top

40 effector genes were selected as metagenes for each cluster, and we scored every cell based on effector genes. For every patient, median value of four meta-cluster scores were used for Pi analysis. We then assigned cells to the four clusters and calculated relations between their proportions and tumor changes. To explore the prognosis value of four meta-clusters in the bulk iCCA RNA-seq samples, we calculated the scores for every sample using GSVA and assigned the sample to the one of four meta-clusters based on the max GSVA score.

### Functional annotation of myeloid subtypes

Markers classifying myeloid functions in Supplementary Data 4 were collected from previous studies including M1, M2, Angiogenesis, Phagocytosis, Anti-inflammatory and Pro-inflammatory[21,35]. Using these markers and the sc.tl.score_genes function from scanpy module, we calculated the functional scores for every myeloid subtype. Mean values of the functional scores were used to evaluate the function of every myeloid subtype[21,35].

### TCR clonal analysis

We applied Scirpy to calculate the clonotypes for all T cells, where each clonotype was defined as clusters of cells with identical complementarity-determining regions 3 (CDR3) nucleotide sequences[81]. Cells with no immune-receptor (IR) identified were excluded for downstream TCR analysis while cells with dual IR were kept, taking the increasing evidence of dual IR T cells population into consideration[82,83]. The number of T cell subtypes in each TCR clone ID was counted to demonstrate the concordance between T cell phenotypes and clonotypes. To confirm whether such concordance remained at patient level, we mapped the top 5 abundant clone IDs with T cell subtypes for each patient.

By comparing the TCR clonotype changes between pre- and post-GOLP samples, we defined 3 types of TCR clones: unique, clonal, and shared. For each patient, the unique clonotype referred to T cells with unique clone IDs while the clonal clonotype referred to T cells with identical clone ID in either pre- or post- sample. Clonotypes that remained unchanged after GOLP treatment were labeled as 'shared'. For paired samples of each patient, the proportions of three types of clones were calculated for CD8+ T cells and CD4+ T cells respectively.

The change in population of TCR clonotypes pre- and post- post GOLP was used as categorical parameter was tested by Fisher exact test. The clonotypes that were significantly reduced post treatment were defined as contracted (Fisher's exact test $P < 0.05$, Odds Ratio > 1) and the clonotypes that were significantly increased or appeared only in the post-GOLP were defined as expanded or novel ($P < 0.05$, Odds Ratio <1). Clonotypes with non-significant $P$-values were defined as 'unchange'. Totally, 318 clonal types were found to show significant change, including 129 contracted, 129 novel and 60 expanded clones. T cells were further labeled by defined TCR clonal change status. According to the expression of newly labeled cell types, we found the novel and expanded clones showed high similarity, and the novel and expanded clone types were thus combined and relabeled as 'expanded'. We used scirpy.tl.alpha_diversity function to calculate the diversity of TCR clonotypes and specified D50 as the metric. D50 was defined as the minimum percentage of distinct CDR3s accounting for at least half of the total CDR3s in a population or subpopulation of immune system cells (Patent number: WO2012097374A1).

### Lineage trajectory analysis of CD8+ T cells

To explore the differential states of CD8+ T cell subtypes and their trajectory differences upon GOLP treatment, we used Monocle2 to infer the lineage trajectory of CD8+ T cells[44]. According to the Moncle2 workflow, a 'CellDataSet' was created and estimated size factors as well as dispersions. Differentially expressed genes (DEGs) were derived by 'differentialGeneTest' function from each cell cluster and genes with a q-value < 0.01 were kept and ranked. Top 400 DEGs were used to order cells in pseudo-time analysis. After the cell trajectories were constructed, DEGs along the pseudo-timeline were detected using the same function. For pseudotime visualization, different types of group factors were used, including cell clusters, pre- and post- GOLP sample. Besides, the exhaustion score of each cell was added to the dataset, and visualized along the lineage trajectory.

### Exhausted and cytotoxic score of T cells

Markers related to cytotoxic and exhausted status of T cells were collected from previous studies[84]. Cytotoxic markers were *PRF1, IFNG, GNLY, NKG7, GZMK, GZMB, GZMA, CST7, TNFSF10*, while exhausted markers were *CTLA4, HAVCR2, LAG3, PDCD1, TIGIT*. We used sc.tl.score_genes to score every T cell. For every T cell subtype, exhausted score density and cytotoxic score density were calculated and smoothed between pre- and post-GOLP samples. Cell clusters barely expressing these markers were selected as controls. We selected CD4 SOCS3+, CD4 naive as non-exhausted cells and CD4+ T cells as non-cytotoxic cells. Their score distribution was applied as statistical background. Using one-tail test at 99% level, exhausted score cutoff and cytotoxic score cutoff were obtained and labeled as grey dash line in the corresponding plots.

### RNA velocity analysis

To explore the possibility of cell transition among CD8+ subtypes, we combined velocyto and scVelo to calculate and visualize the ratio of spliced/unspliced transcripts in each CD8+ T cell subtype to infer their maturation state[85]. In detail, we used run10X subcommand from velocyto to interpret cellranger outs and produced velocyto loom files for each sample. These samples were concatenated to produce a merged velocity h5ad file where splice/unspliced information is stored in multiple layers. The T-cell subtype annotation were inferred from scanpy analyzed data[86]. Using raw matrix as an input, 58% of our single-cell transcriptome data were unspliced and the rest were spliced, with no cells in ambiguous splicing state. The stochastically optimistic model was used to estimate the RNA velocity of each cell in all T cell subtypes. From there, the velocity-inferred directionality formed the basis of the extended PAGA trajectory analysis, which was embedded in the scanpy module as an alternative to traditional pseudotime packages. Here, we used PAGA mapping to velocity to visualize the transition between CD8 Proliferating, CD8 GZMB+, and CD8 GZMK+ subtypes.

### Cell-cell interaction

To investigate the possible mechanisms behind the observed changes, we used NATMI and CellChat for the ligand-receptor pair analyzes in the 3 interested CD8+ T cell subtypes, all myeloid subtypes, and tumor cells which were divided into the aforementioned four classes[47,87]. Compared to CellphoneDB, NATMI helped us to focus on the specifically expressed ligand-receptor pairs. The specificity weight was defined as Eq. 5,

$$\text{edge(celltype1 to celltype2)}_{ligand1-receptor1}^{specificity} = \frac{celltype1_{ligand1}^{mean}}{\sum\left(celltype_{ligand1}^{mean}\right)} \times \frac{celltype2_{receptor1}^{mean}}{\sum\left(celltype_{receptor1}^{mean}\right)} \quad (5)$$

To exclude generic (housekeeping) ligand receptor interactions, we set the expression threshold as 10CPM, specificity threshold of 0.1 and ligand/receptor detection rate as 20%. Simultaneously, the CellChat package was used to predict cell-cell interaction in collective pathway modules. We used the robust mean (trimean) expression per cell cluster for the inference of cell-cell communication network and focused on secreted signals. The prediction from both modules were used in combination to explain our results.

## Transcription factor analysis

To further explore the cell signaling network, we used pySCENIC module to infer the transcription factors regulating each sub populations in CD8+ T cells[88]. Using the expression matrix as an input and lambert 2018 transcription factor list[89] as a reference, we calculated the adjacency scores in the estimated co-expression network. We then used the hg38_refseq-r80_500bp_up_and_100bp_down_tss and hg38_refseq-r80_10kb_up_and_down_tss files in cisTarget public dataset as references for motif enrichment and subsequent regulons deduction. The over-presentation of each motif was estimated by their area under the curve (AUC) scores. Only motifs with normalized enrichment score (NES) ≥ 3.0 and regulons which were either directly annotated or inferred from > 10 orthologue genes were kept, as recommended by the SCENIC documents.

## Evaluation of cluster score and cell-cell interaction validation in iCCA bulk RNA sequencing samples

To evaluate the level of Macro CD5L+ and CD8 GZMB+/GZMK+/proliferating, and to explore their relation, bulk iCCA RNA-seq samples were used. For the representative of Macro CD5L+ in bulk RNA-seq, five markers (CD5L, SLC40A1, FCGR3A, MARCO and SEPP1) were used based on our scRNA-seq clustering makers. For CD8 GZMK+, four markers (GZMK, GZMA, ITM2C, TNFSF9) were used. For CD8 GZMB+, four markers (GZMB, RGS1, RBPJ, CTSD) were used. For CD8 proliferating, four markers (STMN1, TYMS, MKI67, TUBB) were used. The expression level of marker genes for each cell type were averaged to represent the level of the cell type in each bulk RNA-seq sample and designated as score for the cell type. Then a linear regression model was built between the averaged cell type scores for the two cell types in the iCCA cohort. The ratio of these scores was used for survival analysis with the median as a split. The cell-cell interactions were then calculated by their score using single value linear regression.

## Survival and cox analysis

Survival data and corresponding expression data were collected from the bulk-seq analyzes group. We used OS as an indicator of effectiveness. Using R package 'survminer', we created a standard survival table. We applied log-rank test to calculate the survival difference between groups and plotted the survival graphs using 'ggsurvplot' function from survminer. Cox model was built for the four meta-clusters in the bulk iCCA RNA-seq samples using 'coxph' function and further plotted by 'ggforest' function from survminer.

## Flow cytometry

Isolated single-cell suspensions from the tissue were centrifuged (350 × $g$, 6 min, 4°C) and resuspended in 100 µl Cell Staining Buffer (Biolegend, Cat.420201, without dilution). Human TruStain FcX™ (Fc Receptor Blocking Solution, BioLegend Cat. No. 422301, dilution 1:200) was used to incubate with the single-cell suspensions for 15 min at room temperature before cell-surface staining.

For cell-surface staining, the antibodies (APC/PercP-Cy5.5 anti-human CD8 (Biolegend #344721/344709, dilution 1:200); BV421 anti-human CD233(LAG-3) (Biolegend # 369313, dilution 1:100); BV421 anti-human CD161 (Biolegend #339913, dilution 1:100); BV421 anti-human CD152(CTLA4) (Biolegend #369605, dilution 1:100); PE anti-human CD279 (PD-1) (Biolegend # 135205, dilution 1:100); Alexa Fluor® 700/FITC anti-human CD45 (Biolegend #368514/103107, dilution 1:200); APC/PE anti-human FOLR2 (Biolegend #391705/391703, dilution 1:100); APC/PE anti-mouse Folate Receptor β (FR-β) (Biolegend #153305/153303, dilution 1:100); PerCP anti-mouse F4/80 (Biolegend #123125, dilution 1:200); FITC anti-mouse CD45 (Biolegend #157607, dilution 1:200)) were added to incubate with the cell suspensions for 30 min in the dark at 4°C. Then, washed with 2 mL of Cell Staining Buffer by centrifugation for 2 times (350 × $g$, 6 min, 4°C). For intracellular staining, added 1 mL of 1× FOXP3 Fix/Perm solution (Biolegend, Cat.421401, without dilution) to each tube; vortexed and incubated at room temperature in the dark for 20 min, then spined down the cells and removed the supernatant. Washed once with cell staining buffer (BioLegend, Cat. 420201, without dilution) by spin at 300 × $g$ for 6 min and removed the supernatant. Washed once with 1 mL 1× FOXP3 Perm buffer (BioLegend, Cat.421402, without dilution). Re-suspended cells in 1 mL 1× FOXP3 Perm buffer, incubated at room temperature in the dark for 15 min, spined down cells and discarded the supernatant, then resuspended the pellet in 100 µL of 1× FOXP3 Perm buffer. Added the antibodies (FITC anti-human Granzyme K (Biolegend # 370508, dilution 1:100); Brilliant Violet 510™ anti-human Ki-67 (Biolegend #350518, dilution 1:100); PE anti-human/mouse Granzyme B (Biolegend # 372207, dilution 1:100)) and incubated at room temperature in the dark for 30 min. Washed twice with 2 mL of Cell Staining Buffer by centrifugation (350 × $g$, 6 min, 4°C), and resuspended in 500 µL cell staining buffer then analyzed with flow cytometer BD LSRFortessa™ X-20 Flow Cytometer.

## Multiplex immunofluorescence

Multiplex immunofluorescence was performed by BOND-MAX/BOND-III (Leica). In brief, formalin-fixed paraffin-embedded slides or TMA slides were incubated at 62°C overnight, then performed deparaffinization and continued 5 rounds of staining (Round1: GZMB staining; Round2: CD5L staining; Round3: GZMK staining; Round 4: CD68 staining; Round 5: CD8 staining). For every staining Round, antigen retrieval was first performed by ER1 (For Round1; AR9961-CN, Leica) or ER2 (For Round2,3,4,5; AR9640-CN, Leica) at 100°C for 20 min. Rinsing with TBST for three times was performed between the following steps in the staining round. Blocked endogenous enzymes by Blocking Buffer for 10 min. Incubated with primary antibody at room temperature (RT) at RT for 10 min (Round1: GZMB (1:500 dilutions, 30 min); Round2: CD5L (2 µg/mL dilution, 60 min); Round3: GZMK (1:250 dilutions, 60 min); Round4: CD68 (1:400 dilutions, 60 min); Round5: CD8 (1:300 dilutions, 60 min)). Incubated with HRP secondary antibody (Goat anti-Rabbit poly-HRP for Leca DS9800 staining system) at RT for 10 min. Incubated with Neon TSA fluorescent for dye (Round1: Neon TSA 620, 1:100 dilutions; Round2: Neon TSA 520,1:100 dilutions; Round3: Neon TSA 670,1:100 dilutions; Round4: Neon TSA 440,1:100 dilution; Round5: Neon TSA 570,1:100 dilution). All the detailed information of the primary antibodies is listed in Supplementary Data 9.

After 5 rounds of staining, added anti-fluorescence quenching sealing tablets with DAPI for 10 min and then performed imaging by Pannoramic MIDI (3DHISTECH Ltd.). The quantitative analysis was performed by HALO (Indica Labs).

## Reporting summary

Further information on research design is available in the Nature Portfolio Reporting Summary linked to this article.

## Data availability

Raw sequencing data of scRNA and TCR generated in this study have been deposited in the National Genomics Data Center with the accession code (PRJCA021882), with a copy deposition in biosino NODE database (OEP003206). Data are available upon request through the repository portal. The raw sequencing data is only available for non-commercial and academic purposes under controlled access due to ethical and legal restrictions. The corresponding authors will respond to requests for the data in two weeks. The data will be available for three months once access has been granted. The publicly available data of bulkRNA and clinicopathologic information used in this study are available in the Supplemental information (Table S1) of the paper [https://www.cell.com/cancer-cell/fulltext/S1535-6108(21)00659-0#supplementaryMaterial][69]. The cisTarget Human motif database used by SCENIC method[88] in this study are available in

https://resources.aertslab.org/cistarget/motif2tf/motifs-v9-nr.flybase-m0.001-o0.0.tbl. The remaining data are available within the Article, Supplementary Information or Source Data file. Source data are provided with this paper.

## Code availability

The custom code used in this study are available on https://github.com/zhaodalv/GOLP_code.

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

## Acknowledgements

This study was supported by the Clinical Research Plan of SHDC (SHDC2020CR1003A, For Fan Jia), the Program of Shanghai Academic Research Leader (22XD1402700, For Shi Guo-Ming), the National Natural Science Foundation of China (81972232, For Shi Guo-Ming), the Key Disease Joint Research Program of Xuhui District (XHLHGG202102, For Shi Guo-Ming), Sanming Project of Medicine in Shenzhen (SZSM202003009, For Fan Jia), the National Innovation and Entrepreneurship Training Program for College Students (202210246001 S, For Lu Jia-Cheng) and Youth Fund of Zhongshan Hospital Affiliated to Fudan University (For Lu Jia-Cheng). We thank Prof. Yongzhong Liu for providing the cell line AY-LTC2. We thank the Medical Science Data Center in Shanghai Medical College of Fudan University for the data analysis support. We also thank Genergy Bio-Technology (Shanghai) Co. and SeekGene BioSciences (Beijing) Co. for their help in our single-cell sequencing.

## Author contributions

Conceptualization: J.F., J.Z., G.-M.S. Data collection: J.F., J.Z., G.-M.S., X.-Y.H., J.-C.L., C.G., X.-J.G., H.-Y.Z., X.-D.Q., Y.-Z.P.,, X.-L.M., Y.-M.Z., P.-F.Z., J.-B.C., Z.-B.D., N.R., C.H., S.-J.Q., Q.G., Q.-M.S., Y.-H.S., A.-W.K., G.-H.Y., X.-Y.W., Y.C., D.W. Data analysis: Y.D.S., L.-L.W., J.-C.L., Y.-N.S., X.-Y.H., C.G., C.L. Investigation: J.-C.L., L.-L.W., Y.-N.S., X.-Y.H., C.G. Supervision: J.F., J.Z., G.-M.S., Y.-D.S. Visualization: J.-C.L., L.-L.W., Y.-N.S. Writing – original draft: J.-C.L., Y.-N.S., L.-L.W. Writing – review & editing: J.F., G.-M.S., Y.-D.S., X.-Y.H.

## Competing interests

The authors declare no competing interests.

## Additional information

[1]Department of Liver Surgery and Transplantation, Zhongshan Hospital, Fudan University, Shanghai 200032, China. [2]Liver cancer Institute, Fudan University, Shanghai 200032, China. [3]Key Laboratory of Carcinogenesis and Cancer Invasion, Ministry of Education of the People's Republic of China, Shanghai 200032, China. [4]Institute of Neuroscience, CAS Center for Excellence in Brain Science and Intelligence Technology, Chinese Academy of Sciences, Shanghai 200031, China. [5]Department of Pathology, Zhongshan Hospital, Fudan University, Shanghai 200032, China. [6]Department of Intervention Radiology, Zhongshan Hospital, Fudan University, Shanghai, China. [7]Department of Radiology, Zhongshan Hospital, Fudan University, Shanghai 200032, China. [8]Clinical Research Unit, Institute of Clinical Science, Zhongshan Hospital of Fudan University, 200032 Shanghai, China. [9]These authors contributed equally: Jia-Cheng Lu, Lei-Lei Wu, Yi-Ning Sun, Xiao-Yong Huang, Chao Gao, Guo-Ming Shi. ✉e-mail: shi.guoming@zs-hospital.sh.cn; zhou.jian@zs-hospital.sh.cn; ydsun@ion.ac.cn; fan.jia@zs-hospital.sh.cn

