## [Peer Review File · Nature Communications]

Macro CD5L+ deteriorates CD8+T cells exhaustion and impairs combination of Gemcitabine-Oxaliplatin-Lenvatinib-anti-PD1 therapy in intrahepatic cholangiocarcinomaReviewers' Comments:

Reviewer #1:

Remarks to the Author:

Lu et al. present in their study "Macro CD5L deteriorates PD1-independent exhaustion in CD8 T and impairs GOLP response in intrahepatic cholangiocarcinoma" how the immune status may influence tumor therapeutic response and how this may determine therapeutic approaches for patients with intrahepatic cholangiocarcinoma (iCCA). The authors analyzed single-cell transcription and TCR profiles of 18 tumor tissues including pre- and post-therapy samples of a high-ORR combinational GOLP (gemcitabine, oxaliplatin, lenvatinib, and anti-PD1 antibody) for iCCA.

While the presented manuscript addresses a very important and up-to-date topic, there are several substantial and partly fundamental concerns regarding methodological and statistical issues. In addition, the statements and conclusions are in parts not backed by the results presented. Especially, the detailed molecular mechanism of macro CD5L is not clear. The analyses are mainly associations, but the detailed function is not revealed.

1. Methodological issues:

A. The number of 18 tumor tissues seems to be too low to clarify the given questions and the given results therefore may not allow valid conclusions.

B. The authors used only one (!) cell line (mIC-22) which is moreover not very well characterized to my knowledge and hard to find in the literature. Results of assays using only one cell line are not capable to produce robust results. At least two and well established / characterized cell lines should be utilized.

C. Human iCCA are currently subdivided into the large-duct and small-duct type, according to WHO, and this separation has important implications regarding tumor biology, molecular alterations (e.g. IDH, FGFR status) and consequently treatment options and patient outcome. However, this manuscript neglects this subtyping completely and therefore lacks a state-of-the-art tumor classification.

D. GOLP treatment presented in the phase II trial (Zhou et al., JCO, 2021) needs further validation in large randomized clinical trials.

2. The manuscript is partly hard to comprehend and therefore should be rewritten in large parts.

3. GOLP should be spelled out in the title because this abbreviation of a very specific treatment is not common knowledge. In addition, it should be "CD8 T cells" instead of "CD8 T" in the title.

4. Too many uncommon abbreviations are used and they are not explained properly (e.g. GZMB).

5. There are several major language problems (e.g. "Despite the encouraging results of targeted therapy and immune checkpoint inhibitors (ICIs) therapy were achieved in patients with the malignancies, the majority of patients with iCCA still gain limited benefits from single-agent treatment owing to low frequency of target gene mutation and microsatellite stability (MSS).").

Authors should employ a native speaker to improve the language quality of the manuscript.

Reviewer #2:

Remarks to the Author:

In this manuscript, the authors have examined the single-cell transcriptional and TCR profiles of 18 tumor tissues pre- and post-therapy of a high-ORR combinational GOLP (gemcitabine, oxaliplatin, lenvatinib, and anti-PD1 antibody) for intrahepatic cholangiocarcinoma (iCCA) u. They found that high proportions of GZMB+ CD8 T cells and proliferating CD8+ cells and a low Macro CD5L+ proportion predicted good responses to GOLP treatment in iCCA, and the crosstalk between CD5L+ macrophage and GZMB+ CD8 T cells impaired GOLP response by upregulating CTLA-4 in T cells, which was reversed by CTLA4 monoclonal antibody. Although the findings were potentially interesting, most of

them are descriptive and no validation by genetic or molecular biology.

Major concerns:

1. In Fig.1F, the first tumor on the left is more than 2 cm in diameter, too large to meet Laboratory Animal Welfare and Ethical requirements.
2. The authors sort FOLR2+ Macro CD5L to co-culture with CD8+ T cells in Figure 7J. They found that Macro CD5L exacerbated the exhaustion of CD8 GZMB. But whether CD8 GZMB exhaustion was caused by CD5L protein alone was unclear. The authors should provide data to support this conclusion.
3. The authors claimed that the four meta-clusters calculated by their scRNA data also exhibited robustness in a larger iCCA 28 cohorts with bulk RNA data (Fig.3H), and the patients with high cluster C1/C3 probably receive benefit from GOLP therapy. Where is the survival or prognosis data?
4. In Fig.4H, the predictive index (Pi) suggest that iCCA patients with high levels of Macro CD5L may be insensitive to GOLP therapy. How about the therapeutic index (Ti) of Macro CD5L?
5. In Fig.4J, the authors claimed that baseline Macro CD5L percentages of the SD and PR groups in three GOLP treatment cohorts is about 50% on average, which is inconsistent with scRNA-seq data.
6. In Fig.S6D, the authors claimed that T cells with expanded clonotypes upon GOLP treatment expressed both exhaustion (HAVCR2, TIGIT, LAG3, and ENTPD1) and activation (IFNG and TNFRSF9) marker genes, which is not the obvious conclusion in picture. Especially compared with contracted TCR clones, the change of activation marker genes are too mild.
7. It's on page 9, line15-16, it was stated that the researchers identified three major myeloid types: macrophages (CD68), monocytes (CD14), and dendritic cells. However, in Fig.S7G and S7H, they used the CD14+ monocyte marker to gate or sort macrophage CD5L+ (FOLR2+). MDSC is common and heterogenous in tumor microenvironment. Is Macro CD5L a subset of MDSC? Please explain.
8. CTLA4 is primarily expressed on CD4 T cells. In Fig.8, the mechanism of CTLA4 monoclonal antibody reversed GOLP response in iCCA may not only target CD8 GZMB, which also target CD4 Treg (negative therapeutic index in Fig.5D). The authors should provide data to support this conclusion.
9. In Fig.8A, the authors set up a murine model to validate the effects of Macro CD5L. However, PBS is not a good control in this experiment, they may choose other types of macrophage instead.
10. In Fig.5, the CD8 T cells were classified into 5 subtypes based on their gene expression profiles: CD8 GZMB, CD8 GZMK, CD8 KLRB1, CD8 Trm and CD8 proliferating. It would be recommended to further annotate T cell types. For example, the CD8 GZMB is very likely to be TEMRA or effector T cells, CD8 GZMK might be TEM, and CD8 proliferating is likely to be exhausted T cells, according to Fig.6J and Fig.5B, and previous publications by Zemin Zhang and so on. Apparently, the effector T cells (CD8 GZMB) were associated with better prognosis.
11. A conceptual confounding also happened here, The authors suggest a transition from CD8 proliferating to CD8 GZMK through CD8 GZMB (Fig.6I and Fig.6P), namely from exhausted T cells to TEM through effector T cells. This is not a correct conclusion. This transition is not likely to happen.

Minor points:

1. In Fig.3D, it's hard to see the intratumoral heterogeneity of tumor cells. In Fig.3E, they clustered tumor cells and identified 4 meta-clusters (C1-C4) with distinct gene expression profiles. Were the tumor cells of each patient distributed in all meta-clusters or mainly in one meta-cluster?
2. The authors described: "considering that Ctl4, but not Lag3 or Klrb1 highly expressed both in mouse and human CD8 GZMB in the presence of Macro CD5L (Figure 7M and 8D)". Where is Figure 7M?
3. The author also performed correlation analysis of Macro CD5L and CD8 GZMB (Fig.7H) as well as the ratio of CD8 GZMK and CD8 proliferating (Fig. 6M) in the iCCA cohort (FU-iCCA, N = 262) with bulk RNA-seq data. How did they perform this deconvolution? There are many algorithms available in the market. The authors had to explain it in the method section.

Reviewer #3:

Remarks to the Author:

The authors show a comprehensive single cell study of 18 tumor tissues, pre and post therapy.

We congratulate the authors for showing CD8 GMZB and GMZK and CD5 Macrophages association with response and perform a few creative analysis.

Unfortunately, many figures and assumptions are not accurate and require to lower the tone of the findings. Additional computational experiments are necessary to clarify the story presented.

While the author claims are not novel, the additional dataset of paired single cell and TCRseq is relevant to the field.

A major re-write and revision of the figures is necessary. The inaccuracies and poor language of the manuscript on its current version include:

1. Title "Macro CD5L" is a not common notation to refer to Macrophages expressing CD5L. Also, most macrophages express CD5L. Perhaps emphasize the unique phenotype described on the paper?

2. Figure 1 A. The treatments should be sorted by response rate, unless the authors want to emphasize the order of treatments due a specific reason?.

3. Figure 1 F. It's unacceptable. Showing statistics as many lines is doesn't look good and it's very confusing for the reader. Boxplots are not design to show this number of comparisons. Here the authors should revise the type of visualization used, perhaps use heatmaps may be more informative. Also, the tissues showed represent only 1 sample. It would be a stronger case if several samples are shown to demonstrate reproducibility.

4. Supplemental figure 1 shows IHC examples, however the quality of the H&E is very poor. Showing the mice is also not very convincing and quite gruesome. More IHC examples and ROIs emphasizing the findings would be more appropriate.

5. Figure 2D is not a validation of the found cellular phenotypes. It's just a correlation between the cell types.

6. Figure 2E barplots are not clear. Both parts of the figure don't add up to 100% and may indicate some errors done during analysis. If correct, the authors need to clarify how either part of the figure is calculated.

7. TCRseq is used to derived a few ratios, however no information about clones that are enriched in either pre or post is shown.

8. IF findings were not very clear, the quality of the figures was very poor.

9. Overall, all figures, results and conclusions need to be carefully revised and clarified. The authors need to emphasize more how the TCRseq findings help supporting their findings. Increase their emphasis on reproducibility and readability.

Point-by-point responses to the Reviewers' comments

Reviewer #1, expertise in iCCA and immunotherapy (Remarks to the Author):

Lu et al. present in their study “Macro CD5L deteriorates PD1-independent exhaustion in CD8 T and impairs GOLP response in intrahepatic cholangiocarcinoma” how the immune status may influence tumor therapeutic response and how this may determine therapeutic approaches for patients with intrahepatic cholangiocarcinoma (iCCA). The authors analyzed single-cell transcription and TCR profiles of 18 tumor tissues including pre- and post-therapy samples of a high-ORR combinational GOLP (gemcitabine, oxaliplatin, lenvatinib, and anti-PD1 antibody) for iCCA.

While the presented manuscript addresses a very important and up-to-date topic, there are several substantial and partly fundamental concerns regarding methodological and statistical issues. In addition, the statements and conclusions are in parts not backed by the results presented. Especially, the detailed molecular mechanism of macro CD5L is not clear. The analyses are mainly associations, but the detailed function is not revealed.

Response: We thank the reviewer for the critical and constructive comments. In the revised manuscript, we have addressed the issues and questions raised by the reviewer, by performing many new experiments and analyses. The content of the revision is summarized above for all reviewers. In particular, we have performed additional experiments and analyses to further validate our major conclusions based on limited number of available samples. We have also repeated the functional experiments using another available cell line and confirmed the robustness of our main findings.

1. Methodological issues:

A. The number of 18 tumor tissues seems to be too low to clarify the given questions and the given results therefore may not allow valid conclusions.

Response: We thank for the reviewer's question. Although we have performed single-cell data analysis based on 18 samples, we have validated our results and major conclusions using two independent large cohorts as well as conducting *in vitro* and *in vivo* experiments, including multi-color immunofluorescence, flow cytometry, bulk RNA sequencing, *in vivo* mouse experiments, single-cell sequencing of mouse tumors after GOLP therapy, and *in vitro* cell co-culture experiments. In response to the reviewers' comments, we have added more experiments to enhance the reliability of our results, which are listed as follows,

- (1) We additionally used another mouse intrahepatic cholangiocarcinoma cell line and validated that GOLP therapy had the best tumor suppressive effect.
- (2) We sorted the Macro CD5L⁺ subpopulation by FACS, performed ELISA and confirmed that the Macro CD5L⁺ subpopulation has the characteristic of high CD5L protein secretion.
- (3) We conducted Macro CD5L⁺ and CD8⁺ T cell co-culture experiments and validated that the CD5L secreted by Macro CD5L⁺ induced CD8 GZMB⁺ to exhibit exhaustion phenotype characterized by high expression of CTLA4.
- (4) We additionally performed *in vivo* experiments using another mouse intrahepatic cholangiocarcinoma cell line and found that the presence of Macro CD5L⁺ led to GOLP therapy resistance.

(5) By *in vivo* mouse experiments, we verified that the anti-CTLA4 antibody could reverse GOLP therapy resistance by affecting CD8⁺ T cells, instead of Treg cells or other CD4⁺ T cells.

In fact, the sample size in this study was limited by the difficulty in obtaining paired samples with GOLP therapy for scRNA-seq analysis. First, GOLP is a new therapy under clinical trial and no large population is available. Second, scRNA-seq requires fresh tissues, but the number of tissues from pre-treatment puncture samples were mostly limited and not sufficient for scRNA-seq experiments.

These additional experiments and analysis have further validated our conclusions from multiple aspects. We have added the limitation of sample size in the Discussion of the revised manuscript (see **Page 21 Line 26-29; Page 22 Line 1-4**).

B. The authors used only one (!) cell line (mIC-22) which is moreover not very well characterized to my knowledge and hard to find in the literature. Results of assays using only one cell line are not capable to produce robust results. At least two and well established / characterized cell lines should be utilized.

Response: We thank for the reviewer's comments. The mouse intrahepatic cholangiocarcinoma cell line mIC-22 used in this study was constructed by our group. The construction method and phenotype validation of this cell line were described in the Methods part as well as in **Figure R1A-B** and **Figure S1B-F** of the revised manuscript. In addition, multiple academic institutions and research groups have asked to obtain this cell line from us for intrahepatic cholangiocarcinoma research, including Sichuan University, Zhejiang University, Xiangya Medical College, etc. Their follow-up researches will also cite this paper.

In response to the reviewer's suggestion, we have obtained another mouse intrahepatic cholangiocarcinoma cell line AY-LTC2, which was driven by Akt and Yap, the same way of construction can be found in the papers^{1,2}. We validated the expression of CK19 by immunohistochemistry and performed *in vivo* experiments to validate the tumorigenicity of this cell line (**Figure R1C** and **Figure S1G-H**). Using the AY-LTC2 cell line, we similarly found that GOLP therapy showed better therapeutic response compared to single or dual therapies (**Figure R2A** and **Figure 1F**), consistent with our previous results in mIC-22 cell line. Using the subcutaneous model constructed with AY-LTC2 cells, we also found that the presence of Macro CD5L⁺ cells led to mouse resistance to the GOLP therapy (**Figure R2B** and **Figure S8E**). These results together validated our results and suggested the robustness of this study.

Figure R1. Validation of the two mouse iCCA cell line mIC-22 and AY-LTC2
 (A and B) The construction method and phenotype validation of mouse iCCA cell line mIC-22
 (C) The phenotype validation of mouse iCCA cell line AY-LTC2

A

B

Figure 2. Validation of the responses to different treatment in the two mouse iCCA cell lines
 (A) Photos and statistic results of the tumor size of the two mouse iCCA cell lines after GOLP or other single/dual therapies.

(B) Statistic results of the tumor size of the two mouse iCCA cell lines after Macro CD5L⁺ or PBS intratumoral injection and GOLP therapy.

C. Human iCCA are currently subdivided into the large-duct and small-duct type, according to WHO, and this separation has important implications regarding tumor biology, molecular alterations (e.g. IDH, FGFR status) and consequently treatment options and patient outcome. However, this manuscript neglects this subtyping completely and therefore lacks a state-of-the-art tumor classification.

Response: We appreciate for the reviewer's valuable suggestion. We have explored whether the iCCA subtypes had different responses to GOLP treatment. Among the 55 patients receiving GOLP therapy in this study, 24 patients were diagnosed as large-duct and 31 as small-duct subtype. With GOLP therapy, 15 and 16 patients showed PR in the large-duct and small-duct subtypes, respectively. No significant difference was found in GOLP responses between the two groups by

Chi-square test (Table R1 and Table S1), suggesting this subtyping had little impact on the responses to GOLP therapy. Nevertheless, we have added discussion regarding the relationship between the subtype of iCCA and the response of GOLP treatment in the revised manuscript (see Page 6, Line 17-20).

	Small-duct type	Large-duct type
PR	16	15
SD	15	9

Chi-square test $P = 0.4194$

Table R1. Correlation between the two subtypes of iCCA and GOLP response

D. GOLP treatment presented in the phase II trial (Zhou et al., JCO, 2021) needs further validation in large randomized clinical trials.

Response: We fully agree with the reviewer's suggestion that this therapeutic regimen needs further validation in further clinical trials. Despite that the GOLP treatment presented was only phase II results, the therapy has been recommended as a first-line treatment for advanced intrahepatic cholangiocarcinoma in the 2021 Chinese CSCO guidelines due to its significant objective response. In addition, we have currently started a multicenter, double-blinded, randomized, phase III study to confirm the high efficacy of this combination therapy in patients with advanced iCCA, and obtained NMPA approval (No. 2021LP01825). The trial is also registered at Clinicaltrials.gov (No. NCT05342194). We will report the results of this phase III trial in a timely manner after completion. In response to the reviewer's comment, we have added discussions about this issue in the limitations part of the revised manuscript (see Page 21, Line 26-29; Page 22, Line 1-4).

2. The manuscript is partly hard to comprehend and therefore should be rewritten in large parts.

Response: We apologize for the language problems in the original manuscript. We have had the manuscript professionally edited by a native English speaker from AJE (American Journal Experts). We hope the reviewer will find this revised version more readable and comprehensible.

3. GOLP should be spelled out in the title because this abbreviation of a very specific treatment is not common knowledge. In addition, it should be "CD8 T cells" instead of "CD8 T" in the title.

Response: We thank the reviewer's suggestion and have modified the title to "Macro CD5L⁺ deteriorates PD1-independent exhaustion in CD8⁺ T cells and impairs the combination therapy of gemcitabine, oxaliplatin, lenvatinib and anti-PD1 in iCCA".

4. Too many uncommon abbreviations are used and they are not explained properly (e.g. GZMB).

Response: We greatly appreciate for the reviewer's suggestion and have added the explanation of every abbreviation at the first occurrence. In addition, we have attached a table of abbreviations and their corresponding full names (**Table S8**) in the revised manuscript. Specifically, "CD8 GZMB" refers to a CD8 cell type characterized by high expression of *GZMB*. This naming system has been widely used in single-cell sequencing data analysis. To avoid confusion, we have changed the name of this cell type as "CD8 GZMB⁺" in the revised manuscript.

5. There are several major language problems (e.g. "Despite the encouraging results of targeted therapy and immune checkpoint inhibitors (ICIs) therapy were achieved in patients with the malignancies, the majority of patients with iCCA still gain limited benefits from single-agent treatment owing to low frequency of target gene mutation and microsatellite stability (MSS).").

Authors should employ a native speaker to improve the language quality of the manuscript.

Response: We thank for the reviewer's suggestions. We have had the manuscript professionally edited by a native English speaker from AJE (American Journal Experts) to polish the language. Specifically, this sentence has been revised as: "Despite the encouraging results of targeted therapy and immune checkpoint inhibitor (ICI) therapy achieved in patients with some malignancies, the majority of iCCA patients still gain limited benefits from single-agent treatment due to the low frequency of targetable mutations and a lack of microsatellite instability." (See **Page 3, Line 27-29**).

Reviewer #2, expertise in liver cancer models (Remarks to the Author):

In this manuscript, the authors have examined the single-cell transcriptional and TCR profiles of 18 tumor tissues pre- and post-therapy of a high-ORR combinational GOLP (gemcitabine, oxaliplatin, lenvatinib, and anti-PD1 antibody) for intrahepatic cholangiocarcinoma (iCCA) u. They found that high proportions of GZMB⁺ CD8 T cells and proliferating CD8⁺ cells and a low Macro CD5L⁺ proportion predicted good responses to GOLP treatment in iCCA, and the crosstalk between CD5L⁺ macrophage and GZMB⁺ CD8 T cells impaired GOLP response by upregulating CTLA-4 in T cells, which was reversed by CTLA4 monoclonal antibody. Although the findings were potentially interesting, most of them are descriptive and no validation by genetic or molecular biology.

Major concerns:

1. In Fig.1F, the first tumor on the left is more than 2 cm in diameter, too large to meet Laboratory Animal Welfare and Ethical requirements.

Response: We sincerely apologize for this error. We have repeated the corresponding experiments using 2 cell lines (mIC-22 and AY-LTC2) in the revision. To avoid tumors growing too large in the experiments, we sacrificed the mice on Day 35 (previously on Day 38) in the repeated experiment. In this way, all the tumor diameters were less than 2cm, and the results and conclusions were consistent with those in the original manuscript (**Figure R3 and Figure 1E and Figure S1L**).

FigureR3. Workflow and photos of GOLP and other control groups for the treatment of iCCA-bearing mouse with mIC-22 or AY-LTC2.

2. The authors sort FOLR2+ Macro CD5L to co-culture with CD8+ T cells in Figure 7J. They found that Macro CD5L exacerbated the exhaustion of CD8 GZMB. But whether CD8 GZMB exhaustion was caused by CD5L protein alone was unclear. The authors should provide data to support this conclusion.

Response: We appreciate for the reviewer's suggestion. To address this concern, we have included additional experiments in our revised manuscript. First, we sorted Macro CD5L⁺ and Macro CD5L⁻ cells and performed ELISA experiment to detect the CD5L protein level in the supernatants. We found that supernatants of Macro CD5L⁺ cells had significantly higher levels of CD5L protein compared to that of Macro CD5L⁻ cells (**Figure R4A and Figure S7J**). Second, we cultured the CD8⁺ T cells with conditioned media of Macro CD5L⁻ cells, and found that the addition of CD5L protein significantly increased the expression level of the exhaustion marker gene *CTLA4* of CD8 GZMB⁺ cells (**Figure R4B and Figure S7K**). Together, these new results demonstrated that CD5L protein alone could induce an exhausted phenotype of CD8 GZMB⁺ cells. (See **Page 16, Line 17-21**).

Figure R4. Validation of the secretion of CD5L and its function on CD8 GZMB⁺ T cells
(A) The sorted Macro CD5L⁺ secreted significantly higher levels of CD5L protein than Macro CD5L⁻ in the supernatants (t test).
(B) CD8⁺ T cells were cultured with conditioned media from Macro CD5L⁻ with or without addition of CD5L protein. Addition of CD5L protein significantly increased expression of the exhaustion marker CTLA4 on CD8 GZMB⁺ cells.

3. The authors claimed that the four meta-clusters calculated by their scRNA data also exhibited robustness in a larger iCCA 28 cohorts with bulk RNA data (Fig.3H), and the patients with high cluster C1/C3 probably receive benefit from GOLP therapy. Where is the survival or prognosis data?

Response: We apologize for missing the prognosis information in our original manuscript. We have examined the survival data of the 262 GOLP-naïve patients, and found that patients classified in the C3 cluster showed the worst prognosis compared to those in the C1, C2, or C4 clusters (**Figure R5**). Since this iCCA cohort has not received GOLP treatment, their survival data could not reflect the association with GOLP response. While, the existence of C3 cluster in the large cohort might suggest that patients classified as C3 could probably benefit from the GOLP therapy and thus improve their prognosis.

FigureR5. Survival data of the 262 GOLP-naïve patients grouped by C1, C2, C3, or C4 clusters.

4. In Fig.4H, the predictive index (Pi) suggest that iCCA patients with high levels of Macro CD5L may be insensitive to GOLP therapy. How about the therapeutic index (Ti) of Macro CD5L?

Response: We thank for the reviewer's question. Actually, we have calculated the Ti of Macro CD5L⁺ and found that Ti was not significantly correlated with the tumor size changes ($P = 0.25$). This phenomenon might be caused by the limited number of patients with this cell population in the pre-GOLP group (3 out of 8 patients had zero abundance of this cluster before GOLP therapy, thus the corresponding value of Ti cannot be calculated). To address the reviewer's concern, we have newly performed multiplex immunostaining to examine the changes of Macro CD5L⁺ in the cohort with 55 patients (including 21 paired samples). We performed Ti calculation using this large cohort and found that the abundance of this cell type was associated with unfavorable response to GOLP therapy (Ti = -0.54; $P = 0.012$; **Figure R6**).

FigureR6. Correlation between the change of Macro CD5L⁺ and tumor shrinkage during GOLP therapy in the 21 paired samples

5. In Fig.4J, the authors claimed that baseline Macro CD5L percentages of the SD and PR groups in three GOLP treatment cohorts is about 50% on average, which is inconsistent with scRNA-seq data.

Response: We greatly appreciate the reviewer's question. Actually, this inconsistency was caused by different calculation methods between the single-cell and multiplex immunohistochemistry (mIHC) data. In the single-cell analysis, we calculated the percentage of Macro CD5L⁺ cells among all the myeloid cells (marked by *CD14*, *FCGR3A* and *LYZ*). While for the mIHC data, we calculated the percentage by dividing the number of CD68⁺ macrophages, which represent only a subset of the myeloid cells. To address this question, we re-analyzed the single-cell data using CD68⁺ macrophages as the denominator and found the percentage of Macro CD5L⁺ cells was markedly increased to a level similar to that of the mIHC (**Figure R7**).

FigureR7. The ratio of Macro CD5L⁺ before and after the change of denominator

6. In Fig.S6D, the authors claimed that T cells with expanded clonotypes upon GOLP treatment expressed both exhaustion (HAVCR2, TIGIT, LAG3, and ENTPD1) and activation (IFNG and TNFRSF9) marker genes, which is not the obvious conclusion in picture. Especially compared with contracted TCR clones, the change of activation marker genes are too mild.

Response: We apologize for the unclear description in the original manuscript. In our original main text “T cells with expanded clonotypes upon GOLP treatment expressed both exhaustion (*HAVCR2*, *TIGIT*, *LAG3*, and *ENTPD1*) and activation (*IFNG* and *TNFRSF9*) marker genes (**Figure S6D**)”, we only aimed to show that exhaustion and activation markers were all expressed in T cells with expanded clonotypes, but did not emphasize the change of activation marker genes between T cells with expanded clonotypes and those with contracted clonotypes. Actually, the exhaustion markers were highly expressed in the expanded TCR group, whereas activation markers were highly expressed in the contracted TCR group (**Table R2**). In response to the reviewers’ comments, we have rephrased the sentence as “T cells with expanded and contracted clonotypes upon GOLP treatment expressed exhaustion (*HAVCR2*, *TIGIT*, *LAG3*, and *ENTPD1*) and activation (*IFNG* and *TNFRSF9*) marker genes, respectively.” to avoid misunderstanding. (See **Page 12, Line 18-20**).

Gene Name	Log2 fold change	Adjusted p value	Group
HAVCR2	2.1631083	7.820870452154397e-41	Expanded TCR
LAG3	1.7870142	1.1519439842688317e-79	Expanded TCR
ENTPD1	1.3866527	1.954664636930935e-09	Expanded TCR
IFNG	1.8620054	2.1129895432897895e-101	Contracted TCR

TNFRSF9	1.2986175	2.0903263056676478e-15	Contracted TCR
---------	-----------	------------------------	----------------

Table R2. Statistical test result of exhaustion and activation genes between Expanded and Contracted TCR group

7. It's on page 9, line15-16, it was stated that the researchers identified three major myeloid types: macrophages (CD68), monocytes (CD14), and dendritic cells. However, in Fig.S7G and S7H, they used the CD14+ monocyte marker to gate or sort macrophage CD5L+ (FOLR2+). MDSC is common and heterogenous in tumor microenvironment. Is Macro CD5L a subset of MDSC? Please explain.

Response: We appreciate the reviewer for raising this important point. Actually, both CD14 and CD68 were expressed in macrophages and monocytes, and we classified myeloid cells into three major subtypes, with higher expression of CD14 in monocytes and higher CD68 in macrophages. This classification criteria was consistent with previous studies^{3 4}. However, since CD68 is an intracellular marker and could not be used for FACS sorting, we selected the surface protein CD14 to isolate macrophages by flow cytometry, which was also commonly used in cell sorting analysis⁵.

We also confirmed the expression of CD14 in the Macro CD5L⁺ cells using our scRNA-seq data (Figure R8A and Figure S4H). In addition, we examined the expression of known MDSC markers S100A8, S100A9 and S100A12⁶ and found very low expression of these genes in Macro CD5L⁺ cells (Figure R8A and Figure S4H). Moreover, we performed RNA sequencing of the sorted CD45⁺CD14⁺FOLR2⁺ (Macro CD5L⁺) and CD45⁺CD14⁺FOLR2⁻ (Macro CD5L⁻) subpopulations. Differential expression analysis revealed the high expression of CD5L and MRC1/CD206 in the Macro CD5L⁺ cells, whereas MDSC-related genes (S100A8, S100A9 and S100A12) were highly expressed in Macro CD5L⁻ cells (Figure R8B). These results further excluded the possibility of Macro CD5L⁺ as a group of MDSC. (See Page 10, Line 17-20)

FigureR8. The gene expression in Myeloid cells and the sorted Macro CD5L⁺ / Macro CD5L⁻ cells.

(A) UMAP plot showing the expression of CD14, S100A8, S100A9 and S100A12 in Myeloid cells.
 (B) The different genes enriched in the sorted Macro CD5L⁺ / Macro CD5L⁻

8. CTLA4 is primary expressed on CD4 T cells. In Fig.8, the mechanism of CTLA4 monoclonal antibody reversed GOLP response in iCCA may not only target CD8 GZMB, which also target CD4 Treg (negative therapeutic index in Fig.5D). The authors should provide data to support this conclusion.

Response: To address the reviewer's concern, we have performed additional experiments and analyses in the revised manuscript. By analyzing the single cell data from GOLP-treated mice, we found that CTLA4 is expressed not only in CD8 T cells but also in FoxP3⁺ CD4⁺ Treg cells (**Figure R9A and Figure S8G-I**). However, Macro CD5L⁺ cells only induced increased expression of CTLA4 in CD8⁺ T cells, but not in CD4⁺ T cells or Tregs (**Figure R9B; Figure 8D and Figure S8J**). These results suggested that the CTLA4-mediated reversal of GOLP resistance is independent of CD4 Tregs. To further confirm this conclusion, we performed an additional experiment by depleting Tregs using an anti-CD25 antibody⁷, and found that anti-CTLA4 treatment could still reverse Macro CD5L-induced GOLP resistance without the existence of Tregs (**Figure R9C and Figure S8K**). Together, these new results conclusively support that the reversal effect of the treatment regimen was independent of CD4 Treg. (See **Page 18, Line 12-20**)

Figure9. The reversal effect of the treatment regimen was CD4 Treg independent

(A) UMAP of lymphoid cells in mouse iCCA tumor after GOLP therapy and feature plot showing the Ctla4 and Foxp3 expression in mouse lymphoid cells.

(B) The expression level of Ctla4 in the mouse CD4⁺ T cells, Tregs and CD8⁺ T cluster 0,1 (Wilcoxon test).

(C) Difference of tumor mass showed anti-CTLA4 treatment could still reverse Macro CD5L⁺-induced GOLP resistance after depleting Tregs by anti-CD25 antibody.

9. In Fig.8A, the authors set up a murine model to validate the effects of Macro CD5L. However, PBS is not a good control in this experiment, they may choose other types of macrophage instead.

Response: We appreciate the reviewer's suggestion. Following the advice, we have now performed experiments and included Macro CD5L⁻ cells as a negative control. We found that Macro CD5L⁻ treatment did not significantly affect tumor response to GOLP compared to PBS treatment (**Figure R10A and Figure S8D**). By contrast, Macro CD5L⁺ treatment resulted in significantly larger tumors compared to Macro CD5L⁻ treatment, confirming the effects of Macro CD5L⁺ in inducing GOLP resistance. We have also replicated these findings in a second cholangiocarcinoma cell line AY-LTC2 (**Figure R10B and Figure S8E**; see **Page 17, Line 21**)

FigureR10. Tumor size after intra-tumoral injection of Macro CD5L⁻, PBS or Macro CD5L⁺ and treated by GOLP therapy

(A and B) Intra-tumoral injection of Macro CD5L⁻ or PBS showed no significant difference in affecting tumor response to GOLP therapy in mIC-22 or AY-LTC2 model. Macro CD5L⁺ injection resulted in significantly larger tumors mass compared to Macro CD5L⁻ or PBS control in mIC-22 or AY-LTC2 model.

10. In Fig.5, the CD8 T cells were classified into 5 subtypes based on their gene expression profiles: CD8 GZMB, CD8 GZMK, CD8 KLRB1, CD8 Trm and CD8 proliferating. It would be recommended to further annotate T cells types. For example, the CD8 GZMB is very likely to be TEMRA or effector T cells, CD8 GZMK might be TEM, and CD8 proliferating is likely to be exhausted T cells, according to Fig.6J and Fig.5B, and previous publications by Zemin

Zhang and so on. Apparently, the effector T cells (CD8 GZMB) were associated with better prognosis.

Response: We appreciate the reviewer for raising this important point. In response to the reviewer's suggestion, we have retrieved the data from previous publications by Zemin Zhang⁸ and integrated with our single-cell dataset. The integrated analysis revealed that CD8 proliferating, CD8 GZMB⁺ and CD8 GZMK⁺ are similar to the CD8-MKI67 (Proliferative T cells), CD8-PDCD1 (T_{EX}) and CD8-GZMK respectively (**Figure R11 A and B**). Importantly, the expression of MKI67 in CD8⁺ T cells in our paper also confirmed the proliferating characteristic of CD8 proliferating cells (**Figure R11 C**). Additionally, CD8 GZMK⁺ in our paper (the same cluster named CD8-GZMK in the paper by Zemin Zhang⁸) might be T_{EM}, and CD8 KLRB1⁺ in our paper (the same cluster named CD8-SLC4A10 in the paper by Zemin Zhang⁸) might be MAIT based on previous studies by Zemin Zhang's group⁹.

Figure R11. Integration of CD8⁺ T in this paper with the scRNA data of previous publication by Zemin Zhang's group.

- (A) The subclusters of CD8⁺ T in our paper were showed in the UMAP of integrated data.
(B) The subclusters of CD8⁺ T in Zhang's publication were showed in the UMAP of integrated data.
(C) The expression of MKI67 in CD8⁺ T cells in our paper.

11. A conceptual confounding also happened here, The authors suggest a transition from CD8 proliferating to CD8 GZMK through CD8 GZMB (Fig.6I and Fig.6P), namely from exhausted T cells to TEM through effector T cells. This is not a correct conclusion. This transition is not likely to happen.

Response: Following the question above (Please see the response in Question 10), our results have shown that CD8 proliferating is not exhausted CD8⁺ T cells, but proliferative CD8⁺ T cells. CD8 GZMB⁺ is more like exhausted T cell (T_{EX}), and CD8 GZMK⁺ is more like effector memory T cells (T_{EM}). In our study, we found the transition is from CD8 proliferating (the same cluster named CD8-MKI67 (Proliferative T cells) in the paper⁸) to CD8 GZMK⁺ (the same cluster named CD8-GZMK in the paper⁸) through CD8 GZMB⁺ (the same cluster named CD8-PDCD1 (T_{EX}) in the paper⁸). The transmission from CD8 proliferating (CD8-MKI67) to CD8 GZMB (CD8-PDCD1) was also reported in the paper by Zemin Zhang⁸ (**Figure S3E** of *Cell* 179, 829–845, October 31, 2019). Additionally, the transmission from CD8 GZMB⁺ to CD8 GZMK⁺ transmission was also reported¹⁰ (**Figure 7D** of *Sci Immunol.* 2021 Aug 10;6(62): eabg5021). In summary, the transition from CD8 proliferating to CD8 GZMB⁺ and the transition from CD8 GZMB⁺ to CD8 GZMK⁺ are consistent with previous reports^{8,10}.

Minor points:

1. In Fig.3D, it's hard to see the intratumoral heterogeneity of tumor cells. In Fig.3E, they clustered tumor cells and identified 4 meta-clusters (C1-C4) with distinct gene expression profiles. Were the tumor cells of each patient distributed in all meta-clusters or mainly in one meta-cluster?

Response: We thank for the reviewer's question. The tumor cells of each patient were distributed in all meta-clusters (**Figure R12A**) and each meta-cluster contained tumor cells from all the patients (**Figure R12B**).

Figure R12. Distribution of meta-clusters.

(A) The tumor cells of each patient were distributed in all meta-clusters

(B) Each meta-cluster contained tumor cells from all the patients.

2. The authors described: “considering that Ctla4, but not Lag3 or Klrb1 highly expressed both in mouse and human CD8 GZMB in the presence of Macro CD5L (Figure 7M and 8D)”. Where is Figure 7M?

Response: We sincerely apologize for the mistake and have corrected it from **Figure 7M** to **7J** in our revised manuscript.

3. The author also performed correlation analysis of Macro CD5L and CD8 GZMB (Fig.7H) as well as the ratio of CD8 GZMK and CD8 proliferating (Fig. 6M) in the iCCA cohort (FU-iCCA, N = 262) with bulk RNA-seq data. How did they perform this deconvolution? There are many algorithms available in the market. The authors had to explain it in the method section.

Response: We thank for the reviewer's helpful suggestion and have added detailed explanation in the Methods section of the revised manuscript. Briefly, the expression level of marker genes for each cell type were averaged to represent the level of the cell type in each bulk RNA-seq sample and designated as score for the cell type. Then a linear regression model was built between the averaged cell type scores for the two cell types in the iCCA cohort. The ratio of these scores was used for survival analysis with the median as a split.

Reviewer #3, expertise in scRNA-seq and scTCR-seq analysis (Remarks to the Author):

The authors show a comprehensive single cell study of 18 tumor tissues, pre and post therapy. We congratulate the authors for showing CD8 GMZB and GMZK and CD5 Macrophages association with response and perform a few creative analysis.

Unfortunately, many figures and assumptions are not accurate and require to lower the tone of the findings. Additional computational experiments are necessary to clarify the story presented.

While the author claims are not novel, the additional dataset of paired single cell and TCRseq is relevant to the field.

A major re-write and revision of the figures is necessary. The inaccuracies and poor language of the manuscript on its current version include:

1. Title "Macro CD5L" is a not common notation to refer to Macrophages expressing CD5L. Also, most macrophages express CD5L. Perhaps emphasize the unique phenotype described on the paper?

Response: We greatly appreciate the reviewer's comments. Actually, in our scRNA-seq data, only a subgroup of macrophages expressed the CD5L gene and were designated as Macro CD5L (**Figure R13A**). We additionally analyzed an external single cell data of liver cancer⁴, and found that CD5L gene was also expressed in only a subset of macrophages, indicating that CD5L is a specific gene for this subgroup (**Figure R13B**). We have also investigated the functional characteristics of various macrophage subtypes previously (**Figure 4K and Figure R13C**), and found that Macro CD5L exhibited characteristics of M2-like, phagocytic, and anti-inflammatory macrophages. In response to the reviewer's comments, we have revised the name of this cell type as "Macro CD5L⁺" to avoid misunderstanding.

A

B

C

Figure R13. CD5L expression and characteristics of Macro CD5L⁺

(A) Feature Plot showing the expression of CD5L in Myeloid cells in this paper

(B) Feature Plot showing the expression of CD5L in all cells in the external single cell data

(C) Heatmap showing previously defined macrophage signatures across all myeloid subtypes. Macro CD5L⁺ were highlighted as M2-polarized and anti-inflammatory macrophages with mostly phagocytosis-related functions.

2. Figure 1 A. The treatments should be sorted by response rate, unless the authors want to emphasize the order of treatments due a specific reason?

Response: We thank for the reviewer's suggestion and have sorted the treatments by objective response rate (ORR) in the revised manuscript (**Figure R14** and **Figure1A**).

Figure R14. Bar plot showing the overall response rate (ORR) in clinical trials of different first-line therapeutic strategies for iCCA patients.

3. Figure 1 F. It's unacceptable. Showing statistics as many lines is doesnt look good and it's very confusing for the reader. Boxplots are not design to show this number of comparisons. Here the authors should revise the type of visualization used, perhaps use heatmaps may be more informative. Also, the tissues showed represent only 1 sample. It would be a stronger case if several samples are shown to demonstrate reproducibility.

Response: We thank for the reviewer's suggestion. We have revised the **Figure 1F** using heatmaps as suggested (**Figure R15A**). As other Reviewers' suggestion, we repeated the experiments, and have shown all the photos of the tumors in the **Figure R15B** and **Figure S1L** in the revised manuscript. Of note, we have added another mouse intrahepatic cholangiocarcinoma cell line AY-LTC2 to validate the results and conclusions in the revised manuscript.

Figure R15. Tumor size of mIC-22 or AY-LTC2 after GOLP therapy.

(A) Tumor mass after three cycles of different treatments (t test).

(B) Photos of tumor of mIC-22 or AY-LTC2 after different therapy strategies.

4. Supplemental figure 1 shows IHC examples, however the quality of the H&E is very poor. Showing the mice is also not very convincing and quite gruesome. More IHC examples and ROIs emphasizing the findings would be more appropriate.

Response: We apologize for the poor quality of figures in the original manuscript. We have uploaded the figures as TIFF format with higher resolution (≥ 300 PPI) (Figure S1 A, C, F, H) and also generated more IHC examples and ROIs in the Figure R16 and Figure R17.

Figure R16. H&E staining and related ROIs of human iCCA before and after GOLP treatment with higher resolution.

Figure R17. IHC examples and related ROIs of mouse iCCA tumor with higher resolution.

5. Figure 2D is not a validation of the found cellular phenotypes. It's just a correlation between the cell types.

Response: We thank for the reviewer's suggestion and have corrected the description as "Cell type to cell type correlation matrix showed every annotated cellular phenotype was highly correlated

with itself, indicating the uniqueness of expression pattern of the cellular phenotype.” in the revised manuscript (see Page 7, Line 2-3).

6. Figure 2E barplots are not clear. Both parts of the figure dont add up to 100% and may indicate some errors done during analysis. If correct, the authors need to clarify how either part of the figure is calculated.

Response: We apologize for the unclear description in the original manuscript. Actually, for every sample, we divided the cell type into two groups: immune cell types and non-immune cell types. **Figure 2E** left panel showed the percentage of immune cell types and the right panel showed the non-immune cell types. For every sample, immune cell type percentages plus non-immune cell type percentages equal to 100%. Follow the reviewer’s suggestion, we have added more descriptions to clarify the figure: “For every sample, annotated cell types were divided into immune cell type (T cells, NK cells, pDCs, neutrophils, B cells, and plasma cells: Figure 2E left) and non-immune cell type (fibroblasts, epithelial cells, endothelial cells and hepatocytes: Figure 2E right). For every sample, the percentage of immune cell type plus percentage of non-immune cell type is 100%.” (See the figure legend of Figure 2E)

7. TCRseq is used to derived a few ratios, however no information about clones that are enriched in either pre or post is shown.

Response: We thank for the reviewer’s suggestion. The top clone enriched in pre- or post-treatment were shown in Figure R18. We have also provided a table for the enriched clones in pre and post treatment in Table S9.

Figure R18. The clones enriched in pre- or post- treatment.

8. IF findings were not very clear, the quality of the figures was very poor.

Response: We apologize for the inconvenience and have uploaded the figures as TIFF format with high-resolution (≥ 300 PPI) in the revised manuscript.

9. Overall, all figures, results and conclusions need to be carefully revised and clarified. The authors need to emphasize more how the TCRseq findings help supporting their findings. Increase their emphasis on reproducibility and readability.

Response: We thank for the reviewer's suggestion. In the revised manuscript, we have carefully revised the figures, performed additional experiments and analyses, and improved the language by a native English speaker from AJE (American Journal Experts) to address the reviewers' concerns. For the TCR-seq analysis, our findings are listed as below,

First, TCR sequencing helps define distinct TCR changes before and after GOLP therapy. As shown in **Figure 6A**, we categorized TCR clonotypes into four groups based on their fold change after treatment: Novel, Expanded, Unchanged, and Contracted. Based on the classification of TCR clonotype differences, we made the surprising discovery that an increased proportion of TCR expanded subgroups in CD8⁺ T cells indicates a better treatment response, while the proportion of TCR contracted subgroups in CD8⁺ T cells is negatively correlated with treatment response (**Figure 6B**). This finding aligns with recent publications¹¹, indicating that TCR clonal expansion during treatment suggests better response of treatment within the tumor microenvironment.

Second, TCR sequencing reveals that CD8⁺ T cells exhibit a more distinct treatment response compared to CD4⁺ T cells. Our results indicate that the correlation between tumor size change and TCR changes is less pronounced in CD4⁺ T cell than CD8⁺ T cell subgroups, suggesting that CD8⁺ T cells exhibit a more significant treatment response, and their status is more strongly correlated with GOLP response. This provides important evidence for our subsequent focus on CD8⁺ T cells.

Third, TCR sequencing results help us identify three key subgroups associated with treatment response: CD8 GZMB⁺, CD8 GZMK⁺, and CD8 proliferating. Based on TCR sequencing results, we subsequently focused on T-cell subgroups that underwent TCR expansion or contraction, indicating that these subgroups are the most critical T cells directly responding to GOLP therapy. Mapping these expanded/contracted subgroups to our previously defined CD8⁺ T cell subgroups through scRNA-seq revealed that CD8 GZMB⁺, CD8 GZMK⁺, and CD8 proliferating are the most significant treatment response subgroups. This aligns with the results in **Figure 5F**, where these three subgroups showed a direct association with GOLP therapy efficacy, providing a TCR perspective on the direct relevance of these subgroups to efficacy.

Fourth, TCR results help us determine the initial state of CD8⁺ T cells. In TCR clonal diversity calculations, we found that CD8 proliferating has the highest TCR diversity, and T cells with high TCR diversity often exhibit a relatively undifferentiated state¹¹. This provides important evidence for clarifying the relative initial state positions of CD8 proliferating among the three CD8⁺ T cells.

In summary, TCR sequencing results provide crucial evidence and support in identifying CD8⁺ cell types associated with the GOLP treatment and constitute a significant part of our research.

Reference

1. Diggs, L.P., Ruf, B., Ma, C., Heinrich, B., Cui, L., Zhang, Q., McVey, J.C., Wabitsch, S., Heinrich, S., Rosato, U., et al. (2021). CD40-mediated immune cell activation enhances response to anti-PD-1 in murine intrahepatic cholangiocarcinoma. *Journal of Hepatology* 74, 1145–1154. 10/ghnncf.
2. Rizvi, S., Fischbach, S.R., Bronk, S.F., Hirsova, P., Krishnan, A., Dhanasekaran, R., Smadbeck, J.B., Smoot, R.L., Vasmatazis, G., and Gores, G.J. (2017). YAP-associated chromosomal instability and cholangiocarcinoma in mice. *Oncotarget* 9, 5892–5905. 10.18632/oncotarget.23638.
3. Ravindran, A., Dasari, S., Ruan, G.J., Artymiuk, C.J., He, R., Viswanatha, D.S., Abeykoon, J.P., Zanwar, S., Young, J.R., Goyal, G., et al. (2023). Malignant Histiocytosis Comprises a Phenotypic Spectrum That Parallels the Lineage Differentiation of Monocytes, Macrophages, Dendritic Cells, and Langerhans Cells. *Modern Pathology* 36. 10.1016/j.modpat.2023.100268.
4. Xue, R., Zhang, Q., Cao, Q., Kong, R., Xiang, X., Liu, H., Feng, M., Wang, F., Cheng, J., Li, Z., et al. (2022). Liver tumour immune microenvironment subtypes and neutrophil heterogeneity. *Nature*. 10.1038/s41586-022-05400-x.
5. Bian, Z., Gong, Y., Huang, T., Lee, C.Z.W., Bian, L., Bai, Z., Shi, H., Zeng, Y., Liu, C., He, J., et al. (2020). Deciphering human macrophage development at single-cell resolution. *Nature* 582, 571–576. 10.1038/s41586-020-2316-7.
6. Zhao, F., Hoechst, B., Duffy, A., Gamrekelashvili, J., Fioravanti, S., Manns, M.P., Greten, T.F., and Korangy, F. (2012). S100A9 a new marker for monocytic human myeloid-derived suppressor cells. *Immunology* 136, 176–183. 10.1111/j.1365-2567.2012.03566.x.
7. Clemente-Casares, X., Blanco, J., Ambalavanan, P., Yamanouchi, J., Singha, S., Fandos, C., Tsai, S., Wang, J., Garabatos, N., Izquierdo, C., et al. (2016). Expanding antigen-specific regulatory networks to treat autoimmunity. *Nature* 530, 434–440. 10.1038/nature16962.
8. Zhang, Q., He, Y., Luo, N., Patel, S.J., Han, Y., Gao, R., Modak, M., Carotta, S., Haslinger, C., Kind, D., et al. (2019). Landscape and Dynamics of Single Immune Cells in Hepatocellular Carcinoma. *Cell* 179, 829-845.e20. 10.1016/j.cell.2019.10.003.
9. Zhang, L., Yu, X., Zheng, L., Zhang, Y., Li, Y., Fang, Q., Gao, R., Kang, B., Zhang, Q., Huang, J.Y., et al. (2018). Lineage tracking reveals dynamic relationships of T cells in colorectal cancer. *Nature* 564, 268–272. 10.1038/s41586-018-0694-x.
10. Notarbartolo, S., Ranzani, V., Bandera, A., Gruarin, P., Bevilacqua, V., Putignano, A.R., Gobbini, A., Galeota, E., Manara, C., Bombaci, M., et al. (2021). Integrated longitudinal immunophenotypic, transcriptional, and repertoire analyses delineate immune responses in patients with COVID-19. *Science Immunology* 6, eabg5021. 10/gnrg4q.
11. Bassez, A., Vos, H., Van Dyck, L., Floris, G., Arijis, I., Desmedt, C., Boeckx, B., Vanden Bempt, M., Nevelsteen, I., Lambein, K., et al. (2021). A single-cell map of intratumoral changes during anti-PD1 treatment of patients with breast cancer. *Nat Med* 27, 820–832. 10/gjwsxb.

Reviewers' Comments:

Reviewer #1:

None

Reviewer #2:

Remarks to the Author:

This revised manuscript has been improved significantly. However, the following concerns remain and should be addressed by the authors.

1. The author found that addition of CD5L protein can exacerbate the exhaustion of CD8 GZMB in Figure R4B, what is the concentration of CD5L? The concentration of CD5L in serum can reach $\mu\text{g/ml}$ levels, while in Figure R4A, only ng/ml levels of CD5L were detected in the sorted Macro CD5L+ cells in the culture medium.

2. On page 7, line 28-29, the authors mentioned that deletions of chromosomes 3 and 6 and high chromosomal instability were almost observed in all the epithelial cells. But in figure S3A, deletions of chromosome 6 is not obvious. Deletions of chromosomes 1 and 13 seem to be more direct deficiency. Please the authors consider more intuitive visualization methods to show this result.

Reviewer #3:

Remarks to the Author:

Dear Authors,

Thank you for providing additional experiments and results to complement the article's previous weaknesses.

After carefully reviewing all the data provided, I still believe the claims are slightly inflated. However, the article should be accepted for publication.

Well done.

Kind regards

Point-by-point responses to the Reviewers' comments

Reviewer #2 (Remarks to the Author):

This revised manuscript has been improved significantly. However, the following concerns remain and should be addressed by the authors.

Response: We thank the reviewer for the valuable suggestions and have addressed the remaining concerns to improve the quality of our manuscript.

1. The author found that addition of CD5L protein can exacerbate the exhaustion of CD8 GZMB in Figure R4B, what is the concentration of CD5L? The concentration of CD5L in serum can reach $\mu\text{g/ml}$ levels, while in Figure R4A, only ng/ml levels of CD5L were detected in the sorted Macro CD5L⁺ cells in the culture medium.

Response: In the experiment of Figure R4B, we cultured cells for 72h with 1 $\mu\text{g/mL}$ concentration of CD5L. We have added more specific description in the corresponding figure legend of the revised manuscript (Figure legend of Figure S7K; Page 57, Line 11). We sincerely apologize for the mistake in Figure S7J, where the concentration should be $\mu\text{g/mL}$ instead of ng/mL . We have corrected this error in the revised manuscript (Figure S7J).

2. On page 7, line 28-29, the authors mentioned that deletions of chromosomes 3 and 6 and high chromosomal instability were almost observed in all the epithelial cells. But in figure S3A, deletions of chromosome 6 is not obvious. Deletions of chromosomes 1 and 13 seem to be more direct deficiency. Please the authors consider more intuitive visualization methods to show this result.

Response: We greatly appreciate the reviewer for the valuable suggestions. We have further included boxplots showing deletions between epithelial cells and reference cells (fibroblasts and endothelial), and the results better showed that significantly higher deletions of chromosomes 3p and 6q in the epithelial cells (Figure R19A and R19B). By statistical comparison, we found that chromosome 13q and 1p also showed deficiency in the epithelial cells, and chromosome 1q showed amplification in the epithelial cells (Figure R19B). These results were consistent with a previous study reporting deletions in chromosome 6q, 3p and 13q of the cholangiocarcinoma (Homayounfar *et al.*, Hum Pathol, 2009). In the revised manuscript, we have changed the description to "In addition, our inferred CNV variations were highly consistent with a previous reported pattern of chromosomal aberrations in cholangiocarcinoma such as chromosome 3p, 6q, 13q deletions and chromosome 1q amplification" (Page 7, Line 27-29).

A

B

Figure R19. Heatmap and boxplots showing the CNVs between epithelial cells and reference cells.

Reviewer #3 (Remarks to the Author):

Dear Authors,

Thank you for provide additional experiments and results to complement the article's previous weaknesses.

After carefully reviewing all the data provided, I still believe the claims are slightly inflated.

However, the article should be accepted for publication.

Well done.

Kind regards

Response: We thank the reviewer for providing valuable suggestions to improve the quality of our manuscript.

REVIEWERS' COMMENTS

Reviewer #2 (Remarks to the Author):

The authors have fully addressed my previous concerns.

Point-by-point responses to the Reviewers' comments

Reviewer #2 (Remarks to the Author):

The authors have fully addressed my previous concerns.

Response: We thank the reviewer for providing valuable suggestions to improve the quality of our manuscript.